# Remember What You Want to Forget: Algorithms for Machine Unlearning

**Ayush Sekhari**
Cornell University
`as3663@cornell.edu`

**Jayadev Acharya**[*]
Cornell University
`acharya@cornell.edu`

**Gautam Kamath**[*]
University of Waterloo
`g@csail.mit.edu`

**Ananda Theertha Suresh**[*]
Google Research, NY
`theertha@google.com`

## Abstract

We study the problem of unlearning datapoints from a learnt model. The learner first receives a dataset $S$ drawn i.i.d. from an unknown distribution, and outputs a model $\widehat{w}$ that performs well on unseen samples from the same distribution. However, at some point in the future, any training datapoint $z \in S$ can request to be unlearned, thus prompting the learner to modify its output model while still ensuring the same accuracy guarantees. We initiate a rigorous study of generalization in machine unlearning, where the goal is to perform well on previously unseen datapoints. Our focus is on both computational and storage complexity.

For the setting of convex losses, we provide an unlearning algorithm that can unlearn up to $O(n/d^{1/4})$ samples, where $d$ is the problem dimension. In comparison, in general, differentially private learning (which implies unlearning) only guarantees deletion of $O(n/d^{1/2})$ samples. This demonstrates a novel separation between differential privacy and machine unlearning.

## 1 Introduction

Many organizations and companies employ user data to train machine learning models for a wide array of applications, ranging from movie recommendations to health care. While some of these organizations allow users to withdraw their consent from their data being used (at which point the organization will delete the user's data), less savory businesses might covertly retain user data. Given the potential for misuse, legislators worldwide have wisely introduced laws that mandate user data deletion upon request. These include the European Union's General Data Protection Regulation (GDPR), the California Consumer Privacy Act (CCPA), and Canada's proposed Consumer Privacy Protection Act (CPPA).

There is some natural ambiguity present in these guidelines. Is it sufficient to simply delete the user's data, or must one also take action on machine learning systems that used this data for training? Indeed, by now, privacy researchers are well-aware that user data may be extracted from trained machine learning models (e.g., Shokri et al. [2017], Carlini et al. [2019]). In a potentially landmark decision, the Federal Trade Commission recently ordered a company to delete not only data from users who deleted their accounts, but also models and algorithms derived from this data [Federal Trade Commission, 2021]. This suggests that organizations have an obligation to retrain any machine learning models after excluding users whose data has been deleted.

---

[*]Authors in alphabetical order.

However, naïvely retraining models after every deletion request would be prohibitively expensive: training modern machine learning models may take weeks, and use resources of value in the millions. One could instead imagine more careful methods, which attempt to excise the required datapoints from the model: crucially, *without* incurring the cost of retraining from scratch. This notion is called *machine unlearning*. The goal would be to obtain a model which is *identical* to the alternative model that would be obtained when trained on the dataset after removing the points that need to be forgotten. This requirement is rather strong: Ginart et al. [2019] proposed a relaxed notion of deletion, in which the model must only be *close* to the alternative, where closeness is defined in a way reminiscent of differential privacy [Dwork et al., 2006a,b] (our variant of this notion is described in Definition 2). This relaxation has inspired the design of several efficient algorithms for data deletion from machine learning models [Guo et al., 2020, Izzo et al., 2021, Neel et al., 2021, Ullah et al., 2021].

As mentioned before, one naïve strategy involves retraining the model from scratch, sans the deleted datapoints. When the training dataset is large, this approach is undesirable for several reasons. First, it is computationally very expensive. Even iteration over the training data can be too costly, let alone training a new model on it. Second, preserving the entire training dataset consumes a significant amount of storage.[2]

Another straightforward approach involves model checkpointing, in which the learner preemptively stores backup models in which certain points have been excluded. While this strategy makes it easy to quickly return an appropriate backup model upon receiving a deletion request, the downside is that one typically has to store a number of additional models which scales with the training data size, which may be prohibitively large. As we can see from these examples, computational and storage complexity are two vital metrics when designing a machine unlearning algorithm.

Finally, while there has recently been a wealth of results in machine unlearning, all of it has focused on the core problem of *empirical risk minimization*, where the goal is to minimize the training loss. However, to fulfil the promise of machine learning, we desire algorithms that can *generalize* to previously unseen test data. Motivated simultaneously by all of these concerns, our goal is to address the following question:

*How do we design resource-efficient machine unlearning algorithms which generalize?*

**Our contributions.**   We initiate a new line of inquiry in machine unlearning:

- We investigate generalization properties of unlearning algorithms, in particular asking: how many samples can we unlearn while still ensuring good performance on unseen test data? In comparison, prior work focused on the empirical training loss only.

- We consider machine unlearning simultaneously under storage constraints as well as the previously studied computation constraints. Unlike prior work, our algorithms do not require the training data to be available to the unlearning algorithm when deleting samples.

- A clean approach for unlearning is to ignore which particular samples are being unlearnt and directly apply known algorithms and guarantees from differential privacy (DP). We show a strict separation between DP and machine unlearning.

  In particular, algorithms based on DP can delete at most $\widetilde{\Theta}(n/\sqrt{d})$ samples while still retaining test loss performance, where $d$ denotes the dimension of the problem. On the other hand, we provide efficient unlearning algorithms that take into account the particular samples to be unlearnt and show that we can delete up to $\widetilde{O}(n/d^{1/4})$ samples, thus giving a quadratic improvement in terms of dependence of $d$ over DP. Our results apply to both strongly convex and convex loss functions.

## 1.1   Related work

Cao and Yang [2015] introduced the term "machine unlearning," and gave efficient deterministic algorithms for exact unlearning in certain settings. This definition requires an algorithm to have identical outputs on a dataset after deleting a point, and if that point was never inserted. However, their

---

[2]Orthogonal to storage constraints, an additional issue is that government regulations may restrict the learner from storing raw user data for extended periods of time due to privacy concerns. However, we focus on storage as it captures undesirability of a wider range of unsatisfactory solutions.

algorithms are restricted to very structured problems only. Bourtoule et al. [2021] provide unlearning algorithms using a sharding-based strategy, though in a weaker unlearning model (requiring only that it be possible that the output may have arisen), and without error guarantees.

Ginart et al. [2019] introduced the probabilistic notion of unlearning, inspired by differential privacy [Dwork et al., 2006b,a]. Their definition requires the output distribution of the unlearning algorithm to be similar to the output distribution obtained by running the learning algorithm on the dataset without the deleted points. Several recent works [Guo et al., 2020, Izzo et al., 2021, Neel et al., 2021, Ullah et al., 2021] provide theoretical error guarantees for various problem settings under this probabilistic notion of unlearning. While our unlearning setup is closely related to that of Ginart et al. [2019] and in the related works, there are two major differences.

First, the prior work focuses on empirical risk minimization [Guo et al., 2020, Izzo et al., 2021, Neel et al., 2021, Ullah et al., 2021]. In their setup, the goal of the unlearning algorithm is to find approximate minimizers of the empirical loss on the remaining training dataset after deleting samples. In comparison, our focus in this paper is on the test loss, and we wish to understand how many samples can be deleted from a learnt model while still ensuring that the updated model performs well on unseen examples (i.e., the generalization error). As we discuss in Section 3.1, the goal of minimizing the training loss is qualitatively different from that of minimizing the test loss.

Second, the prior work focuses exclusively on the computational cost of unlearning, without concern for associated storage requirements. This has led to approaches which involve memory-intensive checkpointing data structures, which enables fast processing of deletion requests, but consumes potentially impractical amounts of storage. In contrast, we are additionally concerned with memory usage, which highlights the drawbacks of such approaches. Unlike prior work, our algorithms do not require the training data to be available to the unlearning algorithm when deleting samples, and only rely on some cheap-to-store data statistics.

The most closely related work to ours is the *certified data removal framework* of Guo et al. [2020] which provides efficient data deletion algorithms for generalized linear models (linear and logistic regression). While our deletion algorithm is similar to the Newton update removal mechanism considered in their work, there are some important technical differences. First, their unlearning setup requires access to the entire training dataset for deleting samples; we do not require this. Second, they provide theoretical guarantees in terms of the norm of the empirical gradient being small after data removal. In comparison, our guarantees are for the test loss. Third, their unlearning definition requires the learning algorithm to be randomized, and this leads to worse performance guarantees due to added noise. In comparison, we do not need to randomize the learning algorithm. Finally, our guarantees hold for arbitrary convex loss functions and are thus broader in scope.

Several other models of unlearning have been considered. Garg et al. [2020] give an alternative perspective on machine unlearning, grounded in cryptography. Other works in this space focus on exploring privacy risks [Chen et al., 2020] and verification [Sommer et al., 2020] in machine unlearning settings. For specific learning models like SVMs, exact unlearning has been considered under the framework of decremental learning [Cauwenberghs and Poggio, 2001, Tveit et al., 2003, Karasuyama and Takeuchi, 2010, Romero et al., 2007]. However, the primary motivation in these works is to use the framework of decremental learning to estimate the leave-one-out error in order to provide generalization guarantees for the learnt model. Finally, there has also been recent empirical and theoretical work in developing definitions and algorithms for machine unlearning with deep neural networks for application domains in computer vision [Du et al., 2019, Golatkar et al., 2020a,b, Nguyen et al., 2020].

## 2 Preliminaries

Let $\mathcal{D}$ be a distribution over an instance space $\mathcal{Z}$ and $\mathcal{W} \subseteq \mathbb{R}^d$ be the parameter space of a hypothesis class. Let $f \colon \mathcal{W} \times \mathcal{Z} \to \mathbb{R}$ be a loss function. The goal is to minimize the test loss population risk (test loss), given by

$$F(w) := \mathbb{E}_{z \sim \mathcal{D}}[f(w, z)], \tag{1}$$

where $f(w, z)$ is the loss of the hypothesis corresponding to $w \in \mathcal{W}$ on the instance $z \in \mathcal{Z}$. Let $F^* = \min_{w \in \mathcal{W}} F(w)$ be the value of this minimum and $w^*$ be a corresponding minimizer. Since the distribution $\mathcal{D}$ is often unknown, we are restricted to rely on samples to find a small test loss model.

Given $S = (z_1, z_2, \ldots, z_n)$, a set of $n$ samples drawn independently from $\mathcal{D}$, standard learning algorithms minimize the empirical loss given by

$$\widehat{F}_n(w) := \frac{1}{n} \sum_{i=1}^{n} f(w, z_i). \tag{2}$$

## 2.1  Learning

Let $A : \mathcal{Z}^n \to \mathcal{W}$ be a learning algorithm that takes the dataset $S$ and returns a hypothesis $A(S) \in \mathcal{W}$. The quality of $A$ is measured in terms of the difference between the population risk of the hypothesis $A(S)$ and the risk of the best hypothesis $w^*$ in $\mathcal{W}$, i.e., the excess risk

$$\mathbb{E}[F(A(S))] - F^*,$$

where the expectation is over the randomness in $A$ and $S$. This gives a natural notion of sample complexity.

**Definition 1** (Sample complexity of learning). *The $\gamma$-sample complexity of a problem is defined as*

$$n_\gamma := \min\{n \mid \exists A \text{ s.t. } \mathbb{E}_{S \sim \mathcal{D}^n}[F(A(S))] - F^* \le \gamma \quad \text{for all } \mathcal{D}\},$$

*the fewest number of samples with which a $\gamma$-suboptimal minimizer of the population loss $F(w)$ can be achieved for any distribution over the data samples.*

For comparing different algorithms throughout the paper, we set $\gamma = 0.01$ (or any other small arbitrary constant), and require that the provided learning algorithms learn with population risk sub-optimality of at most $0.01$. Standard results in learning theory [Bubeck, 2014, Theorem 6.1] show that for convex and strongly convex losses,

$$n_{0.01} = O(1), \tag{3}$$

where the hidden constant depends on the properties of $f$ such as its Lipschitzness, but is independent of the dimension $d$ of the parameter space $\mathcal{W} \subseteq \mathbb{R}^d$.

## 2.2  Unlearning

Suppose a learning algorithm $A$ over $S$ outputs the model $A(S)$. An unlearning algorithm $\bar{A}$ takes as input the model $A(S)$ and a set $U \subset S$ of data samples that are to be deleted, and is required to output a new model $\widetilde{w} \in \mathcal{W}$. Besides the set $U$ and the model $A(S)$, the unlearning algorithm $\bar{A}$ can also access some additional data statistics $T(S) \in \mathcal{T}$. We emphasize that the unlearning algorithm does not have access to the entire original dataset $S$ in this resource-constrained setting and hence cannot retrain from scratch.

This set of statistics $T(S)$ captures the additional storage required by the algorithm to support unlearning. Thus, one of our goals is to minimize $|T(S)|$, in particular aiming for memory requirements which are independent of the training data size $n$. This precludes strategies which involve storing and reusing the entire training set, or aggressive model checkpointing. On the other hand, it permits storage of simple statistics such as the empirical mean or average gradient of training data points, which may prove useful when unlearning. At the same time, we are still concerned with our unlearning algorithm's time complexity. This goes hand in hand with the storage complexity: for most natural algorithms, the two are likely to be polynomially related. Augmented by this set of data statistics $T(S)$, an unlearning algorithm is a mapping $\bar{A} : \mathcal{Z}^m \times \mathcal{W} \times \mathcal{T} \to \mathcal{W}$.

To illustrate our unlearning setup, consider the following toy example: Suppose we train a model $\widehat{w}$ with four datapoints $S = \{[0.0, 0.1], [2.0, 2.3], [4.0, 0.5], [3.0, 1.3]\}$. After training the model, the learner only keeps $\widehat{w}$ and a cheap-to-store sufficient statistic $T(S)$, and deletes the dataset $S$ from the memory. At this point the learner does not have access to any datapoints from $S$ anymore. Now, when a delete request comes, the request contains just the sample $U$ to be deleted e.g. $[4.0, 0.5]$. The unlearning algorithm at this point will unlearn this sample, using the information $\widehat{w}$, $T(S)$, and $U = [4.0, 0.5]$ only. Here, note that the data sample $[4.0, 0.5]$ is provided to the unlearning algorithm by the user who owns this data and requests for its deletion.

We now define a notion of unlearning, which is motivated by the definition of differential privacy [Dwork et al., 2006a].

**Definition 2** (($\varepsilon, \delta$)-unlearning). *For all $S$ of size $n$ and delete requests $U \subseteq S$ such that $|U| \le m$, and $W \subseteq \mathcal{W}$, a learning algorithm $A$ and an unlearning algorithm $\bar{A}$ is $(\varepsilon, \delta)$-unlearning if*

$$\Pr\big(\bar{A}(U, A(S), T(S)) \in W\big) \le e^\varepsilon \cdot \Pr\big(\bar{A}(\emptyset, A(S \setminus U), T(S \setminus U)) \in W\big) + \delta,$$

*and*

$$\Pr\big(\bar{A}(\emptyset, A(S \setminus U), T(S \setminus U)) \in W\big) \le e^\varepsilon \cdot \Pr\big(\bar{A}(U, A(S), T(S)) \in W\big) + \delta,$$

*where $\emptyset$ denotes the empty set and $T(S)$ denotes the data statistics available to $\bar{A}$.*

The above states that with high probability, an observer cannot differentiate between the two cases (i) the model is trained on the set $S$ and then a set $U$ of $m$ points are deleted by the unlearning algorithm using statistics $T(S)$ and (ii) the model is trained on the set $S \setminus U$ and no points are deleted thereafter by the unlearning algorithm (using statistics $T(S \setminus U)$). For simplicity, throughout the paper, we assume that $\varepsilon \le 1$.

While being similar, the above notion of unlearning is slightly different from the one considered in Ginart et al. [2019]. Specifically, their definition compares the output of the unlearning algorithm after deleting $m$ samples, to the output of the learning algorithm that only operates on $S \setminus U$. Due to this, they require the learning algorithm to be randomized, even in the situation when there would be no delete requests in the future. Thus, the output of the learning algorithm will suffer a degradation in its performance guarantees due to this added noise. On the other hand, our definition does not require the learning algorithm to be randomized since we only compare the output of the unlearning algorithms in the two scenarios—with and without the delete requests. Furthermore, our definition is more general than that of Ginart et al. [2019], as we can simulate their comparison in our definition by considering the unlearning algorithms for which $\bar{A}$ simply adds noise to the output of $A(S \setminus U)$ when $U = \emptyset$.

Our definition of unlearning leads to the following natural definition of the deletion capacity that formalizes how many samples can be deleted while still ensuring good test loss guarantees.

**Definition 3** (Deletion capacity). *Let $\varepsilon, \delta \ge 0$. Let $S$ be a dataset of size $n$ drawn i.i.d. from $\mathcal{D}$, and let $f(w, z)$ be a loss function. For a pair of learning and unlearning algorithms $A, \bar{A}$ that are $(\varepsilon, \delta)$-unlearning, the deletion capacity $m_{\varepsilon,\delta}^{A,\bar{A}}(d, n)$ is defined as the maximum number of samples $U$ that can be unlearnt, while still ensuring an excess population risk of $0.01$. Specifically,*

$$m_{\varepsilon,\delta}^{A,\bar{A}}(d, n) := \max\Big\{m \mid \mathbb{E}\Big[\max_{U \subseteq S : |U| \le m} F(\bar{A}(U, A(S), T(S))) - F^*\Big] \le 0.01\Big\},$$

*where the expectation above is with respect to $S \sim \mathcal{D}^n$ and output of the algorithms $A$ and $\bar{A}$.*

We are primarily interested in unlearning algorithms for which $T(S)$ is small, and in particular does not grow with the dataset size $n$ (which can potentially be very large).

## 2.3 Unlearning via retraining from scratch

The most naïve yet natural baseline for unlearning is to simply retrain the model from scratch using the remaining data. That is, we let $\bar{A}(U, A(S), T(S)) = A(S \setminus U)$. However, the straightforward method to implement this approach would require us to set $T(S)$ to contain the entire training dataset $S$, and thus $|T(S)| \ge n$. However, recall that, we aim to provide unlearning algorithms for which $T(S)$ is independent of $n$. Furthermore, retraining from scratch is computationally expensive – merely reading all the data takes $\Omega(n - m)$ time, not accounting for the cost of actually running the algorithm. These drawbacks lead one to explore more efficient methods for unlearning.

## 3 Our results

Prior works consider unlearning from an optimization perspective, focusing on minimizing the empirical risk, and do not discuss the implications of unlearning on test loss. As we show in the next section, the two could significantly different objectives even for some of the simplest learning problems.

### 3.1 Population risk vs empirical training risk

We first provide a simple example motivating our study of population risk over empirical risk, quantified rigorously in Theorem 1. Consider the following mean estimation problem. Let $d = 1$, $\mathcal{Z} = \mathbb{R}$, and the loss function $f(w, z) = (w - z)^2$. The empirical risk of $n$ points $z_1, z_2, \ldots, z_n$ is minimized by the average $\frac{1}{n} \sum_{i=1}^n z_i$. For this problem there is a simple unlearning algorithm that minimizes empirical risk and also unlearns $U$ exactly. We store the average of the points $w_1 = \sum_{i=1}^n z_i / n$, and upon receiving a deletion request of a set $U$ of $m$ samples, subtract those samples and renormalize to compute the minimizer of the empirical loss on the remaining training samples, i.e., we output

$$w_2 = \frac{n}{n - m} \Big( w_1 - \frac{1}{n} \sum_{z \in U} z \Big). \tag{4}$$

The above update rule requires $T(S)$ to be of size $O(d)$ and completely deletes the samples $U$ satisfying the unlearning guarantee with $\varepsilon = \delta = 0$. The returned solution $w_2$ is the exact minimizer of the empirical loss on left over data points.

However, $w_2$ may not perform well on fresh samples drawn from the test distribution. Consider the same setting as above, but where the points $z_i$ are drawn i.i.d. from Bernoulli(1/2). Thus, the optimal parameter $w^*$ that minimizes the test loss is given by $1/2$. However, consider the scenario where all of the $m$ delete requests correspond to points with value 1. In this case, the minimizer of the updated empirical loss would be smaller by an additive factor of $m/n$ than the previous estimate, and would thus have worse test loss.

In other words, in the unlearning framework, while minimizing training loss might be a good algorithm, just providing guarantees on the training loss can be vacuous if we want the model to generalize to unseen test samples. Furthermore, focusing on the test loss inherently limits the deletion capacity as noted in example discussed above. We further formalize this intuition in Theorem 1 and show that even if the unlearning algorithm has access to all undeleted samples $S \setminus U$, there is a non-trivial limit on the deletion capacity.

**Theorem 1.** *Let $\delta \leq 0.005$ and $\varepsilon = 1$. There exists a 4-Lipschitz, 1-strongly convex function $f$ and distribution $\mathcal{D}$, such that for any learning algorithm $A$ and unlearning algorithm $\bar{A}$, which even has access to undeleted samples $S \setminus U$, the deletion capacity*

$$m_{\varepsilon, \delta}^{A, \bar{A}}(d, n) \leq cn,$$

*where $c$ depends on the properties of function $f$ and is strictly less than 1.*

Prior work [Shalev-Shwartz et al., 2009a, Feldman, 2016] suggests that even in the learning setting there are problems for which empirical risk minimizer solution fails to perform well on fresh samples, and regularization does not help [Kale et al., 2021]. Our situation is worse, since the delete requests $U$ can be adversarially chosen from $S$ [Lai et al., 2016, Diakonikolas et al., 2019]. In fact, the proof of Theorem 1 relies on showing the existence of an adversary that can only delete points and can change the empirical loss considerably. We defer full details to Appendix B.1.

### 3.2 Strict separation between unlearning and differential privacy

Given the strong resemblance between differential privacy and our definition for unlearning, a natural approach would be to use tools from differential privacy (DP) for machine unlearning. The simplest way is to ignore the particular set of delete requests $U$ and construct an unlearning algorithm $\bar{A}$ that only depends on the learning algorithm $A(S)$. More formally, such an unlearning algorithm is of the form $\bar{A}(U, A(S), T(S)) = \bar{A}(A(S))$ and satisfies:

$$\Pr\big(\bar{A}(A(S)) \in W\big) \leq e^{\varepsilon} \Pr\big(\bar{A}(A(S \setminus U)) \in W\big) + \delta,$$
$$\Pr\big(\bar{A}(A(S \setminus U)) \in W\big) \leq e^{\varepsilon} \Pr\big(\bar{A}(A(S)) \in W\big) + \delta.$$

Note that any such pair of algorithms would be differentially private with respect to the original dataset $S$, where the notion of neighboring datasets is for datasets with edit distance of $m$. The above guarantee is stronger than the distribution-free unlearning guarantee in Definition 2, and thus it suffices to satisfy it. The key observation is that any DP algorithm $A$, which is private for datasets with edit distance $m$, automatically unlearns any $m$ data samples. Thus, the standard performance guarantees for DP learning yields the following bound on deletion capacity:

**Lemma 1** (Unlearning via DP)**.** There exists a polynomial time learning algorithm $A$ and unlearning algorithm $\bar{A}$ of the form $\bar{A}(U, A(S), T(S)) = A(S)$ such that the deletion capacity

$$m_{\varepsilon,\delta}^{A,\bar{A}}(d,n) \geq \widetilde{\Omega}\Big(\frac{n\varepsilon}{\sqrt{d\log(e^{\varepsilon}/\delta)}}\Big), \tag{5}$$

where the constant in the $\Omega$-notation above depends on the properties of the loss function $f$.

The above result raises an immediate question of whether this particular dependence on $d$ and $n$ is necessary on the deletion capacity, and if it can be improved further. The following lemma shows that there exist problem instances for which any unlearning algorithm that ignores the samples $U$, can not improve over the factor of $\sqrt{d}$ in the denominator of the deletion capacity bound in (5).

**Lemma 2** (Bassily et al. [2019], Section C)**.** For any learning algorithm $A$ and an unlearning algorithm $\bar{A}$ that does not use $U$, i.e., $\bar{A}(U, A(S)) = \bar{A}(A(S))$, there exists a 1-strongly convex function and $O(1)$-Lipschitz loss function $f$, and a distribution $\mathcal{D}$ such that we can not unlearn even a single sample point, if

$$n \leq c \cdot \frac{\sqrt{d}}{\varepsilon},$$

where $c$ depends on the properties of the function $f$.

Given that the $\sqrt{d}$ dependence on dimension is unavoidable for algorithms that directly use DP, it is natural to wonder whether this factor may be bypassed using other techniques. Our main contribution in this work a positive answer in this direction. As we show in the next section, when the loss function is convex, there exist unlearning algorithms which can delete up to $n/d^{1/4}$ sample points while still retaining the performance guarantee with respect to the test loss.

### 3.3 Unlearning for convex loss functions

In this section, we provide an unlearning algorithm $\bar{A}$ for convex losses that can delete more points than unlearning algorithms that simply use DP.

**Theorem 2.** *There exists a learning algorithm $A$ and an unlearning algorithm $\bar{A}$ such that for any convex (and hence strongly convex), L-Lipschitz, and M-Hessian-Lipschitz loss $f$ and distribution $\mathcal{D}$,*

$$m_{\varepsilon,\delta}^{A,\bar{A}}(d,n) \geq c \cdot \frac{n\sqrt{\varepsilon}}{(d\log(1/\delta))^{1/4}},$$

*where the constant $c$ depends on the Lipschitz constants $L$ and $M$. Furthermore, for the unlearning algorithm $\bar{A}$ has running time $O(d^{\omega})$ where $\omega \in [2, 2.38]$ is the exponent of matrix multiplication, and space complexity for $T(S) = O(d^2)$.*

Theorem 2 and Lemma 2 together show that there exist problem settings, where the deletion capacity in unlearning and DP is different by a multiplicative $\Theta(d^{1/4})$. In particular, when learning with convex loss functions, we can delete $O(n/d^{1/4})$ samples while still retaining good performance on the unseen test loss, whereas DP only guarantees deletion of $\Theta(n/d^{1/2})$ samples. Hence, our algorithm is at least quadratically better in terms of dependence on $d$ in deletion capacity that standard DP algorithms. Besides improving the dependence on $d$, our algorithm also enjoys better dependence on $\varepsilon$ and $\log(1/\delta)$ in the deletion capacity, by at least a quadratic factor.

Our learning algorithm stores additional statistics of the dataset $S$ in order to delete the set $U$ in the unlearning algorithm. The extra memory we need for these statistics is independent of $n$. Furthermore, our algorithm uses the samples in $U$ during the unlearning phase. This paradox of storing and using information in order to delete it, motivates the name of the paper: *Remember what you want to forget*.

Characterizing the entire set of problems for which unlearning and differential privacy are different remains an interesting open question. Theorem 2 yields an improved upper bound, but it is not clear if this dependence on $d$ or $n$ in the deletion capacity is tight or if it can be improved even further. Resolving this question would be a fascinating future research direction.

# 4  Unlearning algorithms

In the following, we provide learning and unlearning algorithms when the loss function $f(\cdot, z)$ is $\lambda$-strongly convex. The unlearning algorithms for convex losses follows by appealing to the strongly convex case after adding regularization. We defer the algorithms and proofs for the convex case to Appendix D. Throughout this section, we make the following assumption:

**Assumption 1.** For any $z \in \mathcal{Z}$, the function $f(w, z)$ is $\lambda$-strongly convex, $L$-Lipschitz and $M$-Hessian Lipschitz with respect to $w$.

**Learning algorithm.**   We denote our learning algorithm by $A_{sc}$. When given a dataset $S$ of $n$ points sampled i.i.d. from some distribution $\mathcal{D}$, the algorithm $A_{sc}$ computes the point $\widehat{w}$ by minimizing the empirical loss $\widehat{F}_n(w)$, i.e.

$$\widehat{w} \leftarrow \operatorname{argmin} \widehat{F}_n(w) := \frac{1}{n} \sum_{z \in S} f(w, z). \tag{6}$$

$A_{sc}$ then returns the point $\widehat{w}$ and the statistics $T(S) := \{\nabla^2 \widehat{F}(\widehat{w})\}$ containing the Hessian of $\widehat{F}(w)$ evaluated at the output point $\widehat{w}$. We provide the pseudocode for $A_{sc}$ in the appendix.

**Unlearning algorithm.**   We denote our unlearning algorithm by $\bar{A}_{sc}$ and provide the pseudocode in Algorithm 1. $\bar{A}_{sc}$ receives as input the set of delete requests $U$, the point $\widehat{w}$ and the data statistics $T(S)$. Using these inputs, $\bar{A}_{sc}$ first estimates the matrix $\widehat{H}$ that denotes the Hessian of the empirical function on the dataset $S \setminus U$ when evaluated at the point $\widehat{w}$. Then, $\bar{A}_{sc}$ computes the point $\bar{w}$ by removing the contribution of the deleted points $U$ from $\widehat{w}$ using the update in (8). Finally, $\bar{A}_{sc}$ perturbs $\bar{w}$ with noise $\nu$ drawn from $\mathcal{N}(0, \sigma^2 \mathbb{I}_d)$ and returns the perturbed point $\widetilde{w}$.

---

**Algorithm 1** Unlearning algorithm ($\bar{A}_{sc}$)

---

**Input:** Delete requests: $U = \{z_j\}_{j=1}^m \subseteq S$, output of $A_{sc}(S)$: $\widehat{w}$, additional statistic $T(S)$ : $\{\nabla^2 \widehat{F}(\widehat{w})\}$, loss function: $f(w, z)$.
1: Set $\gamma = \frac{2Mm^2L^2}{\lambda^3 n^2}$, $\sigma = \frac{\gamma}{\varepsilon}\sqrt{2\ln(1.25/\delta)}$.
2: Compute

$$\widehat{H} = \frac{1}{n - m}\left(n\nabla^2 \widehat{F}(\widehat{w}) - \sum_{z \in U} \nabla^2 f(\widehat{w}, z)\right). \tag{7}$$

3: Define

$$\bar{w} = \widehat{w} + \frac{1}{n - m}(\widehat{H})^{-1} \sum_{z \in U} \nabla f(\widehat{w}, u). \tag{8}$$

4: Sample $\nu \in \mathbb{R}^d$ from $\mathcal{N}(0, \sigma^2 \mathbb{I}_d)$.
5: **Return** $\widetilde{w} := \bar{w} + \nu$.

---

Our main technical insight that leads to improvements in deletion capacity over differential privacy is the following observation. For loss functions that satisfy Assumption 1, when deleting $m$ samples, we can approximate the empirical minimizer on the dataset $S \setminus U$ up to a precision of $O(m^2/n^2)$ by the point $\bar{w}$ computed in (8). This implies that we only need to add noise of the scale of $\sigma \propto O(m^2/n^2)$ to get the desired unlearning guarantee. This noise is smaller than the amount of noise typically added for DP learning [Dwork and Roth, 2014] by a quadratic factor; hence giving us a quadratic improvement in the deletion capacity.

**Lemma 3.** Suppose the loss function $f$ satisfies Assumption 1. Let $S \sim \mathcal{D}^n$ be a set of $n$ samples, and $U \subseteq S$ denote the set of $m$ delete requests. Define the point $\widehat{w}'$ as the empirical minimizer over $S \setminus U$, i.e. $\widehat{w}' \in \operatorname{argmin}_w \sum_{z \in S \setminus U} f(w, z)/(n - m)$. Then,

$$\|\widehat{w}' - \bar{w}\| \leq \frac{2ML^2m^2}{\lambda^3 n^2},$$

where the point $\bar{w}$ is defined in (8) in Algorithm 1.

The following theorem provides performance guarantees for the algorithms $A_{sc}$ and $\bar{A}_{sc}$, and show that $A_{sc}$ and $\bar{A}_{sc}$ are $(\varepsilon, \delta)$-unlearning.

**Theorem 3.** *Suppose the loss function $f$ satisfy Assumption 1 and let the dataset $S \sim \mathcal{D}^n$. Then,*

$(a)$ *The point $\widehat{w}$ returned by running $A_{sc}$ on $S$ satisfies*

$$\mathbb{E}_{S \sim \mathcal{D}^n}[F(\widehat{w}) - \min_{w \in \mathcal{W}} F(w)] \leq \frac{4L^2}{\lambda n}. \tag{9}$$

$(b)$ *For any set $U \subseteq S$ of $m$ delete requests, the point $\widetilde{w}$ returned by $\bar{A}_{sc}$ satisfies*

$$\mathbb{E}_{S,\nu}[F(\widetilde{w}) - \min_{w \in \mathcal{W}} F(w)] = O\Big(\frac{\sqrt{d}Mm^2L^3}{\lambda^3 n^2 \varepsilon}\sqrt{\ln(1/\delta)} + \frac{4mL^2}{\lambda n}\Big). \tag{10}$$

$(c)$ *The learning algorithm $A_{sc}$ and the unlearning algorithm $\bar{A}_{sc}$ are $(\varepsilon, \delta)$-unlearning.*

Note that our guarantees in the above theorem are in terms of the test loss performance, while our algorithms compute minimizers of the training loss. We defer the proof to the Appendix C.2. This gives a lower bound on the number of samples $m$ that can be deleted while still ensuring the desired excess risk guarantee (deletion capacity). Specifically, from the performance guarantee for $\bar{A}_{sc}$ in (10), we observe that we can delete

$$m \geq c \cdot \frac{n\sqrt{\varepsilon}}{(d\log(1/\delta))^{1/4}},$$

samples from the set $S$ (with size $n$) while still ensuring that an excess risk guarantee of $\gamma = 0.01$. Here, $c$ depends on the constants $M, K$ and $\lambda$ for the function $f$. This proves Theorem 2 for strongly convex loss functions.

**Memory.** We do not need to store the entire dataset $S$ for the unlearning algorithm $\bar{A}_{sc}$. We note that the data statistic $T(S)$ that is passed as an input to $\bar{A}_{sc}$ is given by $T(S) = \{\nabla^2 \widehat{F}(\widehat{w})\}$. Clearly, $T(S)$ needs $O(d^2)$ memory and thus $|T(S)|$ is independent of $n$ or $m$.

**Computation.** For the sake of exposition above, our learning algorithm $A_{sc}$ computes the exact minimizer for the empirical loss in (6). However, as we show in Appendix C.2, our theoretical guarantees hold even when the empirical minimizer is computed approximately up to a precision of $O(1/n^2)$. When the domain $\mathcal{W}$ is convex, such a minimizer can be efficiently computed using standard optimization algorithms like accelerated gradient descent, SGD, clipped-SGD, etc. For example, for the $\lambda$-strongly convex case, Nesterov's accelerated GD algorithm can find a $O(1/n^2)$ approximate minimizer in time $\widetilde{O}(nd/\sqrt{\lambda})$ [Bubeck, 2014, Nemirovski and Yudin, 1983]. Furthermore, $A_{sc}$ takes $O(nd^2)$ time to compute $T(S)$.

On the other hand, the running time for the unlearning algorithm $\bar{A}_{sc}$ scales as $O(md^2 + d^\omega)$, the time taken to compute the matrix $\widehat{H}$ and to invert it. Here, $\omega \in [2, 2.38]$. Note that our unlearning time is independent of the (potentially large) dataset size $n$. Furthermore, for problems such as linear SVMs where the Hessian is a diagonal matrix, $A_{sc}$ takes time $O(nd)$ and $\bar{A}_{sc}$ takes time $O(d)$.

**Comparison to retraining from scratch.** Retaining from scratch requires access to the entire training dataset $S$ during unlearning, and is thus not a feasible unlearning algorithm in the resource constrained setting that we consider in this work where the size of the data statistics $T(S)$ should not scale with the size of $S$. However, ignoring the memory issues, there are regimes for the value of $m$ for which retraining from scratch is computationally more efficient that implementing Algorithm 1, and the returned solution enjoys the same test loss performance guarantee. In particular, when $\max(m^5d^2, m^{4+\omega}) > \lambda^4 n^2 \varepsilon$, retraining from scratch by solving (6) using stochastic gradient descent is more efficient that computing the matrix $\widehat{H}$ and its inverse in Algorithm 1. On the other hand, for large datasets (when $n \gg d$), we expect our algorithm to be computationally more efficient. We defer the exact characterization of memory / computation tradeoffs in machine unlearning for future work.

**Algorithms for convex losses.** Our unlearning algorithms when the loss function is convex are based on reductions to the strongly convex setting discussed above. Give the convex loss function $f(\cdot, z)$, we define the function $\widetilde{f}(\cdot, z)$ as

$$\widetilde{f}(w, z) = f(w, z) + \frac{\lambda}{2}\|w\|^2.$$

The key observation is that the function $\widetilde{f}(w, z)$ is $\lambda$-strongly convex, $(L + \lambda\|w\|)$-Lipschitz, $(H + \lambda)$-smooth and $M$-Hessian Lipschitz, and thus we can run algorithms $A_{sc}$ and $\bar{A}_{sc}$ on $\widetilde{f}$. We defer the algorithmic implementation and theoretical analysis for the convex loss setting to Appendix D.

## 5 Conclusion

We initiated a new study on machine unlearning with a focus on population risk minimization, in comparison to previous works that focus on empirical risk minimization. For the case of convex loss functions, we provide a new unlearning algorithm that improves over the deletion capacity, by at least a quadratic factor in $d$, than using an out of the box differentially private algorithm for unlearning. Proving a dimension dependent information theoretic lower bound on the deletion capacity is an interesting future research direction. Another exciting direction of future research is to provide efficient unlearning algorithms for finite / discrete hypothesis class, and for non-convex loss functions. Finally, in this work, we considered the problem of batch deletion where the delete request $U$ all arrive at the same time; Extending our algorithms for the online case is another interesting research direction that we are excited to pursue in future research.

**Acknowledgements**

We thank Robert Kleinberg, Mehryar Mohri, and Karthik Sridharan for helpful discussions. JA is supported in part by the grant NSF-CCF-1846300 (CAREER), NSF-CCF-1815893, and a Google Faculty Fellowship. GK is supported by an NSERC Discovery Grant and a University of Waterloo startup grant.

**Funding Transparency Statement**

Funding in direct support of this work: NSF-CCF-1846300 (CAREER), NSF-CCF-1815893, a Google Faculty Fellowship, an NSERC Discovery Grant, and a University of Waterloo startup grant.

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
