## A  Additional Notation

We recall the following standard definitions for the loss function $f(w; z)$.

**Definition 4** (Lipschitzness). *The function $f(w, z)$ is L-Lipschitz in the parameter $w$ if for all $z \in \mathcal{Z}$, and all $w_1, w_2 \in \mathcal{W}$,*

$$|f(w_1, z) - f(w_2, z)| \leq L\|w_1 - w_2\|.$$

**Definition 5** (Strong convexity). *The function $f(w, z)$ is $\lambda$-strongly convex, if for all $z \in \mathcal{Z}$, and all $w_1, w_2 \in \mathcal{W}$,*

$$f(w_1, z) \geq f(w_2, z) + \langle \nabla f(w_2, z), w_1 - w_2 \rangle + \frac{\lambda}{2}\|w_1 - w_2\|^2.$$

**Definition 6** (Hessian-Lipschitzness). *The function $f(w, z)$ is said to be $M$-Hessian Lipschitz if for all $z \in \mathcal{Z}$, and all $w_1, w_2 \in \mathcal{W}$,*

$$\|\nabla^2 f(w_1, z) - \nabla^2 f(w_2, z)\| \leq M\|w_1 - w_2\|,$$

*or equivalently, that $\|\nabla^3 f(w, z)\| \leq M$ for all $w$.*

## B  Missing proofs from Section 3

### B.1  Proof of Theorem 1

We first develop some technical results, which we will use to prove Theorem 1. For a distribution $\mathcal{D}$, define $\mu(\mathcal{D}) := \mathbb{E}_{Z \sim \mathcal{D}}[Z]$.

**Lemma 4.** There exists two distributions $\mathcal{D}_1$ and $\mathcal{D}_2$ over $[0, 1]$ such that $\|\mathcal{D}_1 - \mathcal{D}_2\|_1 \leq \alpha$ and $|\mu(\mathcal{D}_1) - \mu(\mathcal{D}_2)| \geq \alpha/2$.

**Proof.** Let $\mathcal{D}_1$ be the uniform distribution over $[0.25 + \alpha/2, 0.75 + \alpha/2]$ and $\mathcal{D}_2$ be the uniform distribution over $[0.25, 0.75]$. We note that $\mathbb{E}_{z \sim \mathcal{D}_1}[z] = 0.5 + \alpha/2$ and $\mathbb{E}_{z \sim \mathcal{D}_2}[z] = 0.5$ and hence $|\mathbb{E}_{z \sim \mathcal{D}_1}[z] - \mathbb{E}_{z \sim \mathcal{D}_2}[z]| = \alpha/2$. However, the $\ell_1$ distance between $\mathcal{D}_1$ and $\mathcal{D}_2$ is bounded by $\alpha$.  $\square$

**Lemma 5.** Given two distributions $\mathcal{D}_1$ and $\mathcal{D}_2$ over the domain $\mathcal{Z}$, let the distribution $\bar{\mathcal{D}}$ be defined such that for any $z \in \mathcal{Z}$, $\bar{\mathcal{D}}(z) \propto \min(\mathcal{D}_1(z), \mathcal{D}_2(z))$. Then, for any $\mathcal{D}_1$ and $\mathcal{D}_2$ such that $\|\mathcal{D}_1 - \mathcal{D}_2\|_1 = \alpha$, and $n$ samples $\{z_i\}_{i=1}^n$ drawn iid from $\mathcal{D}_1$, there exists an adversary that deletes at most $2n\alpha$ samples and outputs the dataset $\{z_j\}_{j=1}^{n'}$ with a distribution $\widetilde{\mathcal{D}}^{n'}$ and $n' \leq n - 2n\alpha$ such that

$$\|\widetilde{\mathcal{D}}^* - \bar{\mathcal{D}}^*\|_1 \leq e^{-n\alpha/6},$$

where the superscript denotes distribution over all sequences over the domain $\mathcal{Z}$.

**Proof.** Let $m = 2n\alpha$. Our proof uses two adversaries $A'$ and $A$. We first define $A'$. Given $n$ samples $\{z_i\}_{i=1}^n$, it deletes sample $z_i$ with probability $\frac{[\mathcal{D}_1(z_i) - \mathcal{D}_2(z_i)]_+}{\mathcal{D}_1(z_i)}$, where $[x]_+ := \max\{x, 0\}$. First observe that for any sample $z$, probability that $z$ is retained is given by

$$\mathcal{D}_1(z) \cdot \frac{\min\{\mathcal{D}_1(z), \mathcal{D}_2(z)\}}{\mathcal{D}_1(z)} = \min\{\mathcal{D}_1(z), \mathcal{D}_2(z)\}.$$

Thus the distribution of samples outputted by $A'$ is exactly $\bar{\mathcal{D}}$. However, it can delete more than $m$ samples. Next, we consider another adversary $A$, which is same as $A'$, except it stops after deleting $m$ samples. Hence, for sequences of length $> n - m$, the output of $A'$ and $A$ are the same. Hence,

$$\|\widetilde{\mathcal{D}}^* - \bar{\mathcal{D}}^*\|_1 \leq \Pr(N \geq m), \tag{11}$$

where $N$ is the number of deleted samples. In the rest of the proof, we bound this probability.

Let $X_i \in \{0, 1\}$ be the random variable that takes the value 1 if the sample $i$ is deleted. Hence $N = \sum_i X_i$. Furthermore, $\mathbb{E}[X_i] = \int_z [\mathcal{D}_1(z), \mathcal{D}_2(z)]_+ dz = \|\mathcal{D}_1 - \mathcal{D}_2\|_2/2 = m/4n$. By the Chernoff bound, we have that

$$\Pr\left(\sum_i X_i \geq m\right) \leq e^{-m/12}.$$

Using the above with (11) implies the desired statement.  $\square$

We now have all the tools to prove Theorem 1.

**Proof of Theorem 1.** The proof for small values of $m$ follows from known bounds for sample complexity of learning [Bubeck, 2014, Shalev-Shwartz and Ben-David, 2014]. In the following, we provide an information-theoretic lower bound for $m \geq 100$ by constructing two distributions $\mathcal{D}_1$ and $\mathcal{D}_2$ and show that no single learning-unlearning pair $A, \bar{A}$ can perform well on both of them.

Let $\mathcal{W} = [0, 1]$ and $\mathcal{Z} = R$. Further, let the loss function be $f(w, z) = (w - z)^2$. Our proof consists of two main parts: we first provide a reduction from learning to mean estimation, and then give the lower bound by a reduction from mean estimation to hypothesis testing.

**Reduction from learning to mean estimation:** We first show that for any distribution $\mathcal{D}$ for the population loss given by $F(w) := \mathbb{E}_{z \sim D}[f(wz)]$ satisfies,

$$F(w) - F(w^*) = (w - w^*)^2.$$

To observe this note that

$$
\begin{aligned}
F(w) &= \mathbb{E}[(w - z)^2] \\
&= \mathbb{E}[(w - w^* + w^* - z)^2] \\
&= \mathbb{E}[(w - w^*)^2] + 2\,\mathbb{E}[(w - w^*)(w^* - z)] + \mathbb{E}[(w^* - z)^2] \\
&= (w - w^*)^2 + 2(w - w^*)\,\mathbb{E}[(w^* - z)] + F(w^*) \\
&= (w - w^*)^2 + F(w^*),
\end{aligned}
$$

where the last inequality uses the fact that $\mathbb{E}[z] = w^*$ for our loss function. Thus, in order to bound the learning error, it suffices to bound the error in estimating the mean of the underlying distribution $w^*$.

**From mean estimation to the lower bound:** Since $\mathcal{Z}$ is unbounded and having more information only helps, we assume that the unlearner has access to the entire sample set $S$ i.e., the passed data statistics $T(S) = \{S\}$. Since the output $A(S)$ can be derived from $S$, it suffices to consider unlearning algorithms $\bar{A}$ of the form $\bar{A}(U, S)$. Let $w$ denote the output $\bar{A}(U, S)$. By the definition of forgetting rule,

$$
\begin{aligned}
\mathbb{E}_{\mathcal{D}_1}(\bar{A}(U, A(S)) - w_1^*)^2 &= \int_w \Pr_{\mathcal{D}_1}(\bar{A}(U, A(S)) = w)(w - w_1^*)^2 dw \\
&\geq e^{-\varepsilon} \int_w \Pr_{\mathcal{D}_1}(\bar{A}(\phi, A(S \setminus U)) = w)(w - w_1^*)^2 dw - \delta,
\end{aligned}
$$

where the last inequality uses the fact that $\mathcal{W} = [0, 1]$. Let $\tilde{A} = \bar{A}(\phi, A(\cdot))$. Let $\alpha = m/2n$ and let $\mathcal{D}_1$ and $\mathcal{D}_2$ be two distributions such that $\|\mathcal{D}_1 - \mathcal{D}_2\|_1 = \alpha$. Let the set of delete requests $U$ be chosen by the adversary in Lemma 5. For these set of samples $U$, we have that

$$\int_w \Pr_{\mathcal{D}_1}(\bar{A}(\phi, A(S \setminus U)) = w)(w - w_1^*)^2 dw \geq \int_w \Pr_{\bar{\mathcal{D}}}(\bar{A}(\phi, A(S \setminus U)) = w)(w - w_1^*)^2 dw - e^{-m/12},$$

where $\bar{\mathcal{D}}$ is defined in Lemma 5. Similarly, we have that

$$\mathbb{E}_{\mathcal{D}_2}(\bar{A}(U, A(S)) - w_1^*)^2 \geq e^{-\varepsilon} \int_w \Pr_{\bar{\mathcal{D}}}(\bar{A}(\phi, A(S \setminus U)) = w)(w - w_1^*)^2 dw - e^{-m/12} - \delta.$$

The sum of errors the unlearner makes on either of the errors is at least

$$
\begin{aligned}
\mathbb{E}_{\mathcal{D}_1}[F(w)] - F_{\mathcal{D}_1}^* &+ \mathbb{E}_{\mathcal{D}_2}[F(w)] - F_{\mathcal{D}_2}^* \\
&\geq e^{-\varepsilon} \int_w (\Pr_{\bar{\mathcal{D}}}(\tilde{A}(S \setminus U) = w)((w - w_1^*)^2 + (w - w_2^*)^2) dw - 2\delta - 2e^{-m/12} \\
&\geq \frac{e^{-\varepsilon}}{2} \int_w (\Pr_{\bar{\mathcal{D}}}(\tilde{A}(S \setminus U) = w)((w_1^* - w_2^*)^2) dw - 2\delta - 2e^{-m/12} \\
&= \frac{e^{-\varepsilon}(w_1^* - w_2^*)^2}{2} - 2e^{-m/12} - 2\delta,
\end{aligned}
\tag{12}
$$

where the inequality in the second last line follows from the fact that $a^2 + b^2 \geq (a+b)^2/2$ for any $a, b \in \mathbb{R}$.

Plugging in the distributions $\mathcal{D}_1$ and $\mathcal{D}_2$ as given in Lemma 4 with $\alpha = m/2n$ in the error bound (12) implies that

$$\mathbb{E}_{\mathcal{D}_1}[F(w)] - F^*_{\mathcal{D}_1} + \mathbb{E}_{\mathcal{D}_2}[F(w)] - F^*_{\mathcal{D}_2} \geq \frac{e^{-\varepsilon}m^2}{32n^2} - 2e^{-m/12} - 2\delta.$$

Since $\delta \leq 0.005$, $\varepsilon \leq 1$ and $m \geq 100$, for the above quantity to be less than $0.01$, we have that $m \leq cn \cdot e^{\varepsilon}$, where $ce^{\varepsilon} < 1$. $\qquad\square$

## B.2 Proof of Lemma 1

**Proof.** We let $A$ to be a DP algorithm which is private for datasets with edit distance $m$. Our unlearning algorithm $\bar{A}$ simply returns the input point $A(S)$ without making any changes to it, i.e. $\bar{A}(U, A(S), T(S)) = A(S)$. Clearly, the unlearning algorithm $\bar{A}$ does not require any additional data statistics and thus $T(S) = \emptyset$.

We set the DP algorithm $A$ as the mini-batch noisy SGD method from Bassily et al. [2019]. The learning guarantee for $A$ from Bassily et al. [2019, Theorem 3.2] together with the group privacy property of differential privacy [Vadhan, 2017, Lemma 2.2] implies that:

$$F(A(S)) - F^* \leq 10BL\Big(\frac{1}{\sqrt{n}} + \frac{m\sqrt{d\log(me^{\varepsilon}/\delta)}}{\varepsilon n}\Big). \tag{13}$$

Furthermore, $A(S)$ is $(\varepsilon, \delta)$-DP for datasets with edit distance $m$, i.e. for any set $U \subseteq S$ of $m$ samples:

$$\Pr(A(S) \in W) \leq e^{\varepsilon}\Pr(A(S \setminus U) \in W) + \delta,$$
$$\Pr(A(S \setminus U) \in W) \leq e^{\varepsilon}\Pr(A(S) \in W) + \delta.$$

Since $T(S) = \emptyset$ and $\bar{A}(A(S), U, T(S)) = A(S)$ for any $U \subseteq S$ such that $|U| = m$, we can rewrite the DP guarantee as:

$$\Pr\big(\bar{A}(U, A(S), T(S)) \in W\big) \leq e^{\varepsilon} \cdot \Pr\big(\bar{A}(\emptyset, A(S \setminus U), T(S \setminus U)) \in W\big) + \delta,$$
$$\Pr\big(\bar{A}(\emptyset, A(S \setminus U), T(S \setminus U)) \in W\big) \leq e^{\varepsilon} \cdot \Pr\big(\bar{A}(U, A(S), T(S)) \in W\big) + \delta,$$

implying that the pair $(A, \bar{A})$ is $(\varepsilon, \delta)$-unlearning for $U$ of size $m$.

We next bound the deletion complexity. The bound in the right hand side of (13) implies that we can delete

$$m = \widetilde{\Omega}\Big(\frac{0.01\varepsilon}{\sqrt{\log(e^{\varepsilon}/\delta)}} \cdot \frac{n}{\sqrt{d}}\Big)$$

samples while still ensuring that the excess risk is bounded by $\gamma = 0.01$. The above implies the desired lower bound on the deletion capacity. $\qquad\square$

## B.3 Proof of Theorem 2

The following lower bound on the deletion capacity is based on the excess risk guarantees for our learning and unlearning algorithms given in Theorem 3 and Theorem 4 (in Appendix D) for strongly convex and convex loss setting respectively.

**Proof.** We consider the strongly convex loss and convex loss setting separately below.

**Strongly convex loss setting.** Our learning algorithm $A_{sc}$ and the unlearning algorithm $\bar{A}_{sc}$ are given in Algorithm 2 and Algorithm 1 respectively. Theorem 3 implies that the learning algorithm $A_{sc}$ and the unlearning algorithm $\bar{A}_{sc}$ are $(\varepsilon, \delta)$-unlearning. Furthermore, we have that

$$\mathbb{E}[F(\widehat{w}) - F^*] \leq \frac{4L^2}{\lambda n},$$

and

$$\mathbb{E}[F(\widetilde{w}) - F^*] = O\Big(\frac{\sqrt{d}Mm^2L^3}{\lambda^3 n^2 \varepsilon}\sqrt{\ln(1/\delta)} + \frac{4mL^2}{\lambda n}\Big),$$

where $\widehat{w}$ denotes the output point $A_{sc}(S)$ and $\widetilde{w}$ denotes the output point $\bar{A}_{sc}(U, A_{sc}(S), T(S))$.

The above upper bound on the excess risk implies that we can delete at least

$$m = c \cdot \frac{n\sqrt{\varepsilon}}{(d\log(1/\delta))^{1/4}},$$

samples while still ensuring an excess risk guarantee of $\gamma = 0.01$. Here, the constant $c$ depends on the constants $M, L$ and $\lambda$ for the function $f$. This gives us the desired lower bound on the deletion capacity $m_{\varepsilon,\delta}^{A_{sc},\bar{A}_{sc}}(d, n)$.

**Convex loss setting.** Our learning algorithm $A_c$ and the unlearning algorithm $\bar{A}_c$ are given in Algorithm 3 and Algorithm 4 respectively. Lemma 13 implies that the learning algorithm $A_c$ and the unlearning algorithm $\bar{A}_c$ are $(\varepsilon, \delta)$-unlearning. Furthermore, as a consequence of Corollary 2, we note that setting $\lambda$ as in (25) implies that:

$$\mathbb{E}[F(\widehat{w}) - F^*] = O\Big(c_1\sqrt{\frac{m}{n}} + c_2\Big(\frac{d\log(1/\delta)}{\varepsilon^2}\Big)^{1/8}\sqrt{\frac{m}{n}}\Big)$$

and

$$\mathbb{E}[F(\widetilde{w}) - F^*] = O\Big(c_1\sqrt{\frac{m}{n}} + c_2\Big(\frac{d\log(1/\delta)}{\varepsilon^2}\Big)^{1/8}\sqrt{\frac{m}{n}}\Big),$$

where $\widehat{w}$ denotes the output point $A_c(S)$ and $\widetilde{w}$ denotes the output point $\bar{A}_c(U, A_c(S), T(S))$, and the constants $c_1$ and $c_2$ depend on the properties of the function $f$.

The above upper bound on the excess risk implies that we can delete at least

$$m = c \cdot \frac{n\sqrt{\varepsilon}}{(d\log(1/\delta))^{1/4}},$$

samples while still ensuring an excess risk guarantee of $\gamma = 0.01$. Here, the constant $c$ depends on the constants $M, L$ and $B$ for the function $f$. This gives us the desired lower bound on the deletion capacity $m_{\varepsilon,\delta}^{A_c,\bar{A}_c}(d, n)$.

$\square$

## C    Missing details from Section 4

### C.1    Proof of Lemma 3

**Lemma 6.** The points $\widehat{w}$ and $\widehat{w}'$, defined in Lemma 3, satisfy the following guarantee

$$\|\widehat{w} - \widehat{w}'\| \le \frac{2mL}{\lambda n}.$$

**Proof.** Define the functions $\widehat{F}_1$ and $\widehat{F}_2$ as

$$\widehat{F}_1(w) := \frac{1}{n}\sum_{z\in S} f(w, z) \qquad \text{and,} \qquad \widehat{F}_2(w) := \frac{1}{n-m}\sum_{z\in\bar{S}} f(w, z),$$

where the set $\bar{S} := S \setminus U$. Note that $\widehat{w} = \operatorname{argmin}_w \widehat{F}_1(w)$ and, $\widehat{w}' = \operatorname{argmin}_w \widehat{F}_2(w)$. We first observe that

$$n\big(\widehat{F}_1(\widehat{w}') - \widehat{F}_1(\widehat{w})\big) = \sum_{z\in S} f(\widehat{w}', z) - \sum_{z\in S} f(\widehat{w}, z)$$

$$= \sum_{z \in \bar{S}} f(\widehat{w}', z) - \sum_{z \in \bar{S}} f(\widehat{w}, z) + \sum_{z \in U} f(\widehat{w}', z) - \sum_{z \in U} f(\widehat{w}, z)$$

$$= (m - n)\big(\widehat{F}_2(\widehat{w}') - \widehat{F}_2(\widehat{w})\big) + \sum_{z \in U} f(\widehat{w}', z) - \sum_{z \in U} f(\widehat{w}, z)$$

$$\overset{(i)}{\leq} \sum_{z \in U} f(\widehat{w}', z) - \sum_{z \in U} f(\widehat{w}, z) \overset{(ii)}{\leq} mL\|\widehat{w}' - \widehat{w}\|, \tag{14}$$

where the equality in the second line above follows from the fact that $\bar{s} = S \setminus U$ and the equality in the third line holds from the definition of the function $\widehat{F}_2$. The inequality $(i)$ holds because $\widehat{w}'$ is the minimizer of the function $\widehat{F}_2(w)$ and the inequality $(ii)$ is due to the fact that the function $f$ is $L$-lipschitz. Next, note that the function $\widehat{F}_1$ is $\lambda$-strongly convex. Thus,

$$\widehat{F}_1(\widehat{w}') - \widehat{F}_1(\widehat{w}) \geq \frac{\lambda}{2}\|\widehat{w} - \widehat{w}'\|^2. \tag{15}$$

Using (14) and (15), we get that

$$\frac{\lambda n}{2}\|\widehat{w} - \widehat{w}'\|^2 \leq mL\|\widehat{w} - \widehat{w}'\|,$$

which implies that $\|\widehat{w} - \widehat{w}'\| \leq \frac{2mL}{\lambda n}$. $\qquad\square$

**Proof of Lemma 3.** Given the function $f$ that satisfies Assumption 1, define the functions $\widehat{F}_1$ and $\widehat{F}_2$ as

$$\widehat{F}_1(w) := \frac{1}{n} \sum_{z \in S} f(w, z) \qquad \text{and,} \qquad \widehat{F}_2(w) := \frac{1}{n - m} \sum_{z \in \bar{S}} f(w, z),$$

where the set $\bar{S} := S \setminus U$. Using the Taylor's expansion for $\nabla \widehat{F}_2(\widehat{w}')$ around the point $\widehat{w}$, we get that

$$\|\nabla \widehat{F}_2(\widehat{w}') - \nabla \widehat{F}_2(\widehat{w}) - \nabla^2 \widehat{F}_2(\widehat{w})[\widehat{w}' - \widehat{w}]\| \leq \frac{M}{2}\|\widehat{w} - \widehat{w}'\|^2,$$

where $M$ denotes the Hessian-Lipschitz constant for the function $f(\cdot, z)$, i.e. $\|\nabla^3 \widehat{F}_2(\widehat{w})\| \leq M$. Since $\widehat{w}'$ is a minimizer of $\widehat{F}_2$, and $\widehat{F}_1$ is smooth, we have that $\nabla \widehat{F}_2(\widehat{w}') = 0$. Plugging this in the above bound, we get

$$\|\nabla \widehat{F}_2(\widehat{w}) + \nabla^2 \widehat{F}_2(\widehat{w})[\widehat{w}' - \widehat{w}]\| \leq \frac{M}{2}\|\widehat{w} - \widehat{w}'\|. \tag{16}$$

Also note that

$$\nabla \widehat{F}_2(\widehat{w}) = \frac{1}{n - m} \sum_{z \in \bar{S}} \nabla f(\widehat{w}, z)$$

$$= \frac{1}{n - m} \sum_{z \in S} \nabla f(\widehat{w}, z) - \frac{1}{n - m} \sum_{z \in U} \nabla f(\widehat{w}, z)$$

$$= \frac{n}{n - m} \nabla \widehat{F}_1(\widehat{w}) - \frac{1}{n - m} \sum_{z \in U} \nabla f(\widehat{w}, z)$$

$$= -\frac{1}{n - m} \sum_{z \in U} \nabla f(\widehat{w}, z)$$

where the equality in the second line above holds because $\bar{S} = S \setminus U$, the third line follows by using the definition of the function $\widehat{F}_1(w)$, and the last line is due to the fact that $\widehat{w}$ is the minimizer for the function $\widehat{F}_1(w)$ and hence $\nabla \widehat{F}_1(\widehat{w}) = 0$. Plugging the above in (16), we get that

$$\left\|-\frac{1}{n - m} \sum_{z \in U} \nabla f(\widehat{w}; u) + \nabla^2 \widehat{F}_2(\widehat{w})[\widehat{w}' - \widehat{w}]\right\| \leq \frac{M}{2}\|\widehat{w} - \widehat{w}'\|^2. \tag{17}$$

---
**Algorithm 2** Learning algorithm ($A_{sc}$)
---
**Input:** Dataset $S : \{z_i\}_{i=1}^n \sim \mathcal{D}^n$, loss function: $f$.
  1: Compute

$$\widehat{w} \leftarrow \operatorname{argmin} \widehat{F}_n(w) := \frac{1}{n} \sum_{i=1}^n f(w, z_i).$$

  2: **Return** $(\widehat{w}, \nabla^2 \widehat{F}(\widehat{w}))$.
---

Now, let us define the vector $v$ such that

$$\widehat{w}' = \widehat{w} + \frac{1}{n-m}(\nabla^2 \widehat{F}_2(\widehat{w}))^{-1} \sum_{z \in u} \nabla f(\widehat{w}; z) + v. \tag{18}$$

Plugging the above relation in (17), we get that

$$\|\nabla^2 \widehat{F}_2(\widehat{w})v\| \leq \frac{M}{2}\|\widehat{w} - \widehat{w}'\|^2. \tag{19}$$

Since, the function $\widehat{F}_2$ is $\lambda$-strongly convex, we have that $\|\nabla^2 \widehat{F}_2(\widehat{w})v\| \geq \lambda\|v\|$ for any vector $v$. Using this fact in (19), we get that

$$\|v\| \leq \frac{M}{2\lambda}\|\widehat{w} - \widehat{w}'\|^2.$$

Finally, an application of Lemma 6 implies that $\|\widehat{w} - \widehat{w}'\| \leq \frac{2mL}{\lambda n}$, using which in the above bound, we get that

$$\|v\| \leq \frac{2Mm^2L^2}{\lambda^3 n^2}.$$

Plugging in the definition of the vector $v$ from (18), we get that

$$\left\|\widehat{w}' - \widehat{w} - \frac{1}{n-m}(\nabla^2 \widehat{F}_2(\widehat{w}))^{-1} \sum_{z \in u} \nabla f(\widehat{w}; z)\right\| \leq \frac{2Mm^2L^2}{\lambda^3 n^2}.$$

The desired bound follows by setting $\widehat{H} := \frac{1}{n-m} \sum_{z \in S \setminus U} \nabla^2 f(\widehat{w}, z) = \nabla^2 \widehat{F}_2(\widehat{w})$. $\qquad\square$

### C.2 Proof of Theorem 3

Before we delve into the proof of Theorem 3, we first provide in Algorithm 2, the pseudocode for the learning algorithm $A_{sc}$. We also recall the following technical result that provides excess risk guarantees for the empirical risk minimizer when the loss is strongly convex and Lipschitz.

**Lemma 7** (Claim 6.2 in Shalev-Shwartz et al. [2009b]). *For any $z \in \mathcal{Z}$, let $f(w, z)$ be a $L$-Lipschitz and $\lambda$-strongly convex function in the variable $w$. Given any distribution $\mathcal{D}$, let $S = \{z_i\}_{i=1}^n$ denote a dataset of $n$ samples drawn independently from $\mathcal{D}$. Let the point $\widehat{w}$ be defined as $\widehat{w} := \operatorname{argmin}_w \frac{1}{n} \sum_{i=1}^n f(w, z_i)$. Then,*

$$\mathbb{E}[F(\widehat{w}) - F(w^*)] \leq \frac{4L^2}{\lambda n},$$

*where the function $F(w) := \mathbb{E}_{z \sim \mathcal{D}}[f(w, z)]$ and $w^* \in \operatorname{argmin}_w F(w)$.*

We are now ready to prove the statements of Theorem 3. We prove each part in a separate lemma below. The following result provides performance guarantee for the output of the learning algorithm $A_{sc}$.

**Lemma 8** (Learning guarantee for $A_{sc}$). *For any distribution $\mathcal{D}$, the output $\widehat{w}$ of running Algorithm 2 on the dataset $S \sim \mathcal{D}^n$ satisfies*

$$\mathbb{E}_{S \sim \mathcal{D}^n}[F(\widehat{w})] - F^* \leq \frac{4L^2}{\lambda n},$$

*where $F^*$ denotes $\min_{w \in \mathcal{W}} F(w)$.*

**Proof of Lemma 8.** We note that the point $\widehat{w}$ is given by the empirical risk minimizer on the dataset $S$, i.e.

$$\widehat{w} \leftarrow \underset{w}{\operatorname{argmin}} \frac{1}{n} \sum_{z \in S} f(w, z).$$

Since the function $f(w, z)$ is $\lambda$-strongly convex and $L$-Lipschitz, the desired performance guarantee for the ERM point follows from Lemma 7. $\square$

Next, we provide performance guarantees for the output of the unlearning algorithm $\bar{A}_{sc}$.

**Lemma 9.** For any dataset $S$, output $\widehat{w}$ of $A_{sc}(S)$ and set $U$ of $m$ delete requests, the point $\widetilde{w}$ returned by Algorithm 1 satisfies

$$\mathbb{E}[F(\widetilde{w}) - F^*] = O\Big(\frac{\sqrt{d} M m^2 L^3}{\lambda^3 n^2 \varepsilon} \sqrt{\ln(1/\delta)} + \frac{4mL^2}{\lambda n}\Big),$$

where the expectation above is taken with respect to the dataset $S$ and noise $\nu$.

**Proof of Lemma 9.** We recall that

$$\widetilde{w} = \widehat{w} + \frac{1}{n-m}(\widehat{H})^{-1} \sum_{z \in u} \nabla f(\widehat{w}, z) + \nu, \tag{20}$$

where the vector $\nu \in \mathbb{R}^d$ is drawn independently from $\mathcal{N}(0, \sigma^2 \mathbb{I}_d)$ with $\sigma$ given by $\sqrt{2 \ln(\frac{1.25}{\delta})} \cdot \frac{2 M m^2 L^2}{\lambda^3 n^2 \varepsilon}$. Thus,

$$\mathbb{E}[F(\widetilde{w}) - F(w^*)] = \mathbb{E}[F(\widetilde{w}) - F(\widehat{w}) + F(\widehat{w}) - F(w^*)]$$

$$= \mathbb{E}[F(\widetilde{w}) - F(\widehat{w})] + \mathbb{E}[F(\widehat{w}) - F(w^*)] \leq \mathbb{E}[L\|\widetilde{w} - \widehat{w}\|] + \frac{4L^2}{\lambda n}, \tag{21}$$

where the inequality in the last line follows from the fact that the function $F = \mathbb{E}[f(w, z)]$ is $L$-Lipschitz, and by using Lemma 8. Further, from the relation in (20), we have that

$$\mathbb{E}[\|\widetilde{w} - \widehat{w}\|] = \mathbb{E}[\|\frac{1}{n-m}(\widehat{H})^{-1} \sum_{z \in U} \nabla f(\widehat{w}, z) + \nu\|]$$

$$\overset{(i)}{\leq} \mathbb{E}[\sum_{z \in U} \|\frac{1}{n-m}(\widehat{H})^{-1} \nabla f(\widehat{w}, z)\|] + \mathbb{E}[\|\nu\|]$$

$$\overset{(ii)}{\leq} \sum_{z \in U} \frac{1}{(n-m)\lambda} \mathbb{E}[\|\nabla f(\widehat{w}, z)\|] + \sqrt{\mathbb{E}[\|\nu\|^2]},$$

where the inequality in $(i)$ follows from an application of the triangle inequality, and the inequality $(ii)$ holds because the function $F(w)$ is $\lambda$-strongly convex which implies that $\nabla^2 F(\widehat{w}) \succcurlyeq \lambda \mathbb{I}_d$, and by an application of Jensen's inequality to bound $\mathbb{E}[\|\nu\|]$. Next, using the fact that the vector $\nu \sim \mathcal{N}(0, \sigma^2 \mathbb{I}_d)$, we get that

$$\mathbb{E}[\|\widetilde{w} - \widehat{w}\|] \leq \frac{1}{(n-m)\lambda} \mathbb{E}[\sum_{z \in U} \|\nabla f(\widehat{w}, z)\|] + \sqrt{d}\sigma$$

$$\leq \frac{mL}{(n-m)\lambda} + \sqrt{d}\sigma,$$

where the last line holds because $f(w, z)$ is $L$-Lipschitz. Using the above bound in (21), we get

$$\mathbb{E}[F(\widetilde{w}) - F(w^*)] \leq \frac{mL^2}{(n-m)\lambda} + \sqrt{d}\sigma L + \frac{4L^2}{\lambda n}.$$

Our final guarantee follows by plugging in the value of $\sigma$ and using the fact that $n = \Omega(m)$. $\square$

Finally, we show that the algorithms $A_{sc}$ and $\bar{A}_{sc}$ are $(\varepsilon, \delta)$-unlearning.

**Lemma 10** (Unlearning guarantee). For any distribution $\mathcal{D}$, dataset $S$ and set of delete requests $U \subseteq S$, the algorithms $A_{sc}$ and $\bar{A}_{sc}$ satisfy the following guarantees for any set $W \subseteq \mathbb{R}^d$,

(a) $\Pr(\bar{A}_{sc}(U, A_{sc}(S), T(S)) \in W) \leq e^\varepsilon \Pr(\bar{A}_{sc}(\emptyset, A_{sc}(S \setminus U), T(S \setminus U)) \in W) + \delta$, and

(b) $\Pr(\bar{A}_{sc}(\emptyset, A_{sc}(S \setminus U), T(S \setminus U)) \in W) \leq e^\varepsilon \Pr(\bar{A}_{sc}(U, A_{sc}(S), T(S)) \in W) + \delta$.

**Proof of Lemma 10.** The proof follows along the lines of the proof of the differential privacy guarantee for the Gaussian mechanism (see e.g., Dwork and Roth [2014, Appendix A]).

Let $\widehat{w}$ denote the output of the learning algorithm $A_{sc}$ when run on dataset $S$, and let $\widetilde{w}$ denote the corresponding output of the unlearning algorithm $\bar{A}_{sc}$ when run with delete requests $U$, the input model $\widehat{w}$ and data statistics $T(S)$, i.e. $\widehat{w} = A_{sc}(S)$ and $\widetilde{w} = \bar{A}_{sc}(U, \widehat{w}, T(S))$. Additionally, let $\bar{w}$ be the local variable defined in line 3 of $\bar{A}_{sc}$ (see Algorithm 1) when computing $\widetilde{w}$.

Similarly, let $\widehat{w}'$ denote the output of the learning algorithm $A_{sc}$ when run on dataset $S \setminus U$, and let $\widetilde{w}'$ denote the corresponding output of the unlearning algorithm $\bar{A}_{sc}$ when run with delete requests $\emptyset$, the input model $\widehat{w}'$ and data statistics $T(S \setminus U)$, i.e. $\widehat{w}' = A_{sc}(S \setminus U)$ and $\widetilde{w}' = \bar{A}_{sc}(\emptyset, \widehat{w}', T(S \setminus U))$. Additionally, let $\bar{w}'$ be the local variable defined in line 3 of $\bar{A}_{sc}$ for this case.

Note that in the algorithm $A_{sc}$, the points $\widehat{w}$ and $\widehat{w}'$ are computed as:

$$\widehat{w} = \operatorname*{argmin}_w \frac{1}{n} \sum_{z \in S} f(w, z) \qquad \text{and} \qquad \widehat{w}' = \operatorname*{argmin}_w \frac{1}{n-m} \sum_{z \in S \setminus U} f(w, z).$$

An application of Lemma 3 thus gives us the bound

$$\left\| \widehat{w}' - \widehat{w} - \frac{1}{n-m} (\widehat{H})^{-1} \sum_{z \in U} \nabla f(\widehat{w}, z) \right\| \leq \frac{2Mm^2L^2}{\lambda^3 n^2},$$

where the matrix $\widehat{H}$ is defined in (7). Using the relation in (8) for the points $\bar{w}$ and $\bar{w}'$, and observing that $\widehat{w}' = \bar{w}'$ since $U = \emptyset$ in the calculation of $\widehat{w}'$, we get that

$$\| \bar{w}' - \bar{w} \| \leq \frac{2Mm^2L^2}{\lambda^3 n^2} =: \gamma. \tag{22}$$

Next, note that in the algorithm $\bar{A}_{sc}$, the points $\widetilde{w}$ and $\widetilde{w}'$ are computed as $\widetilde{w} = \bar{w} + \nu$ and $\widetilde{w}' = \bar{w}' + \nu$ respectively, where the noise $\nu \sim \mathcal{N}(0, \sigma^2 \mathbb{I}_d)$ with $\sigma = (\gamma/\varepsilon) \cdot \sqrt{2 \ln(1.25/\delta)}$. Thus, following the same proof as Dwork and Roth [2014, Theorem A.1], with the bound (22), we get that for any set $W$,

$$\Pr(\widetilde{w} \in W) \leq e^\varepsilon \Pr(\widetilde{w}' \in W) + \delta,$$

and

$$\Pr(\widetilde{w}' \in W) \leq e^\varepsilon \Pr(\widetilde{w} \in W) + \delta,$$

giving us the desired unlearning guarantee. $\qquad \square$

**Proof of Theorem 3.** The statements of the theorem follow immediately from Lemma 8, Lemma 9 and Lemma 10 respectively. $\qquad \square$

# D  Unlearning algorithms for convex loss function

In this section, we provide learning and unlearning algorithms when $f$ is convex (but not necessarily strongly convex). Similar to the strongly convex setting, we assume that

**Assumption 2.** For any $z \in \mathcal{Z}$, the function $f(w, z)$ is convex, $L$-Lipschitz and $M$-Hessian Lipschitz with respect to $w$.

Additionally, we also assume the following about some minimizer of the population loss $F(w) = \mathbb{E}_{z \sim \mathcal{D}}[f(w, z)]$.

**Assumption 3.** There exists a $w^* \in \operatorname{argmin}_{w \in \mathcal{W}} F(w)$ such that $\|w^*\| \leq B$.

---

**Algorithm 3** Learning algorithm ($A_c$)

---

**Input:** Dataset $S : \{z_i\}_{i=1}^n \sim \mathcal{D}^n$, loss function: $f$, regularization parameter: $\lambda$.

1: Define
$$\widetilde{f}(w, z) := f(w, z) + \frac{\lambda}{2}\|w\|^2.$$

2: Run the algorithm $A_{sc}$ on the dataset $S$ with loss function $\widetilde{f}$.

3: **return** $(\widehat{w}, \widetilde{H}) \leftarrow A_{sc}(S; \widetilde{f})$, where $\widetilde{H} := \frac{1}{n}\sum_{z \in S} \widetilde{f}(\widehat{w}, z)$.

---

---

**Algorithm 4** Unlearning algorithm ($\bar{A}_c$)

---

**Input:** Delete requests: $U = \{z_j\}_{j=1}^m \subseteq S$, output of $A_c$: $\widehat{w}$, additional statistic $T(S) : \widetilde{H}$, loss function: $f$, regularization parameter: $\lambda$.

1: Define
$$\widetilde{f}(w, z) := f(w, z) + \frac{\lambda}{2}\|w\|^2.$$

2: Run the algorithm $\bar{A}_{sc}$ for the delete request $U$ with input model $\widetilde{w}$ and with loss function $\widetilde{f}$.

3: **return** $\widetilde{w} \leftarrow \bar{A}_{sc}(S, \widehat{w}, T(S); \widetilde{f})$.

---

Our algorithms for convex losses are based on algorithms for the strongly convex setting. Given the convex function $f(\cdot, z)$, define the function $\widetilde{f}(\cdot, z)$ as

$$\widetilde{f}(w, z) = f(w, z) + \frac{\lambda}{2}\|w\|^2.$$

The key observation is that at any $w \in \mathbb{R}^d$, the function $\widetilde{f}(w, z)$ is $\lambda$-strongly convex, $(L + \lambda\|w\|)$-Lipschitz, $(H + \lambda)$-smooth and $M$-Hessian Lipschitz in $w$ for any $z$. Clearly, the function $\widetilde{f}$ satisfies Assumption 1 whenever $w$ is such that $\|w\| \leq L/\lambda$ (see Lemma 14), and thus we can run algorithms $A_{sc}$ and $\bar{A}_{sc}$ respectively on the function $\widetilde{f}$.

Our learning and unlearning algorithms for the convex loss $f$ simply invoke the algorithms $A_{sc}$ and $\bar{A}_{sc}$ on the function $\widetilde{f}$ with an appropriate choice of $\lambda$. We provide the pseudocode in Algorithm 3 and Algorithm 4 respectively.

In order to avoid confusion in this section, for any algorithm $A$, we use the notation $A(S; f)$ to denote the fact that $A$ is run on the loss function $f$ (similarly for $\bar{A}$). Whenever clear from context, we will drop the argument $f$ from the notation. Additionally, we also define $\widetilde{F}$ to denote the population loss w.r.t. the loss function $\widetilde{f}$, i.e. $\widetilde{F}(w) := \mathbb{E}_{z \sim \mathcal{D}}[\widetilde{f}(w, z)]$.

**Theorem 4.** *Suppose the loss function $f$ satisfy Assumption 2 and Assumption 3. Let the dataset $S \sim \mathcal{D}^n$. Then,*

($a$) *The point $\widehat{w}$ returned by running $A_c$ on $S$ satisfies*

$$\mathbb{E}_{S \sim \mathcal{D}^n}[F(\widehat{w}) - \min_{w \in \mathcal{W}} F(w)] \leq \frac{\lambda B^2}{2} + \frac{16L^2}{\lambda n}. \tag{23}$$

($b$) *For any set $U \subseteq S$ of $m$ delete requests, the point $\widetilde{w}$ returned by $\bar{A}_c$ satisfies*

$$\mathbb{E}_{S,\nu}[F(\widetilde{w}) - \min_{w \in \mathcal{W}} F(w)] = O\Big(\frac{\lambda B^2}{2} + \frac{\sqrt{d}Mm^2L^3}{\lambda^3 n^2 \varepsilon}\sqrt{\ln(1/\delta)} + \frac{mL^2}{\lambda n}\Big). \tag{24}$$

($c$) *The learning algorithm $A_c$ and the unlearning algorithm $\bar{A}_c$ are $(\varepsilon, \delta)$-unlearning.*

**Corollary 1.** Suppose we did not have any unlearning requests, and only cared about the performance of the output point $\widehat{w}$ for the learning algorithm $A_c$. Then, setting $\lambda = L/B\sqrt{n}$, the performance guarantee for the point $\widehat{w}$ given in Theorem 4 implies

$$\mathbb{E}[F(\widehat{w}) - F^*] \leq \frac{BL}{\sqrt{n}}.$$

The above rate is tight for learning with Lipschitz convex losses (see Bubeck [2014, Theorem 6.1]).

**Corollary 2.** Suppose that we have $m$ delete requests and thus care about the performance guarantee of both the point $\widehat{h}$, output of the learning algorithm $A_c$, and the point $\widetilde{w}$, output of the unlearning algorithm $\bar{A}_c$. In this case, we set the regularization parameter $\lambda$ as:

$$\lambda = \max\Big\{\frac{L}{B}\sqrt{\frac{m}{n}}, \Big(\frac{\sqrt{d}Mm^2L^3}{B^2n^2\varepsilon}\sqrt{\ln\big(1/\delta\big)}\Big)^{1/4}\Big\}. \tag{25}$$

Plugging the above values of $\lambda$ in Theorem 4, we get that

$$\mathbb{E}[F(\widehat{w}) - F^*] = O\Big(c_1\sqrt{\frac{m}{n}} + c_2\Big(\frac{d\log(1/\delta)}{\varepsilon^2}\Big)^{1/8}\sqrt{\frac{m}{n}}\Big)$$

and

$$\mathbb{E}[F(\widetilde{w}) - F^*] = O\Big(c_1\sqrt{\frac{m}{n}} + c_2\Big(\frac{d\log(1/\delta)}{\varepsilon^2}\Big)^{1/8}\sqrt{\frac{m}{n}}\Big),$$

where the constant $c_1 \propto BL$ and $c_2 \propto \big(\frac{ML^3}{B^2}\big)^{1/4}$.

## D.1 Proof of Theorem 4

We are now ready to prove the statements of Theorem 4. We prove each part in a separate lemma below. The following provides performance guarantee for the output of the learning algorithm $A_c$.

**Lemma 11** (performance guarantee for $A_c$). For any $\lambda > 0$ and $S \sim \mathcal{D}^n$, the point $\widehat{w}$ returned by Algorithm 3 satisfies

$$\mathbb{E}[F(\widehat{w}) - F^*] \leq \frac{\lambda B^2}{2} + \frac{16L^2}{\lambda n}.$$

**Proof.** First, note that an application of Lemma 14 implies that for any dataset $S$, the empirical minimizer $\widehat{w}$ (returned by $A_c$) satisfies: $\|\widehat{w}\| \leq L/\lambda$. Thus, our domain of interest is $\mathcal{W} := \{w \mid \|w\| \leq L/\lambda\}$. Over the set $\mathcal{W}$, the function $\widetilde{f}$ is $2L$-Lipschitz, and thus, an application of Lemma 8 implies that the returned point $\widehat{w}$ satisfies

$$\mathbb{E}[\widetilde{F}(\widehat{w})] \leq \widetilde{F}(\widetilde{w}^*) + \frac{16L^2}{\lambda n},$$

where $\widetilde{w}^*$ denotes the minimizer of $\widetilde{F}(w)$. We can further upper bound the right hand side above as

$$\mathbb{E}[\widetilde{F}(\widehat{w})] \leq \widetilde{F}(w^*) + \frac{16L^2}{\lambda n},$$

where $w^*$ denotes a minimizer of the population loss $F(w)$ that satisfies Assumption 3. Plugging in the form of the function $\widetilde{F}(w)$ in the above, we get

$$\mathbb{E}[F(\widehat{w})] \leq F(w^*) + \frac{\lambda}{2}\|w^*\|^2 + \frac{16L^2}{\lambda n}$$
$$\leq F(w^*) + \frac{\lambda B^2}{2} + \frac{16L^2}{\lambda n},$$

where the last line holds due to Assumption 3. $\qquad\square$

**Lemma 12** (performance guarantee for $\bar{A}_c$). For any $\lambda > 0$, dataset $S \sim \mathcal{D}^n$, output $\widehat{w}$ of $A_c(S)$ and set $U$ of $m$ delete requests, the point $\widetilde{w}$ returned by Algorithm 4 satisfies

$$\mathbb{E}[F(\widehat{w}) - F^*] = O\Big(\frac{\lambda B^2}{2} + \frac{\sqrt{d}Mm^2L^3}{\lambda^3n^2\varepsilon}\sqrt{\ln(1/\delta)} + \frac{mL^2}{\lambda n}\Big).$$

**Proof.** Let $w^* \in \operatorname{argmin}_w F(w)$. We note that

$$
\begin{aligned}
\mathbb{E}[F(\widetilde{w}) - F(w^*)] &= \mathbb{E}[F(\widetilde{w}) - F(\widehat{w}) + F(\widehat{w}) - F(w^*)] \\
&= \mathbb{E}[F(\widetilde{w}) - F(\widehat{w})] + \mathbb{E}[F(\widehat{w}) - F(w^*)] \\
&\leq \mathbb{E}[L\|\widetilde{w} - \widehat{w}\|] + \mathbb{E}[F(\widehat{w}) - F(w^*)],
\end{aligned}
\tag{26}
$$

where the inequality in the last line holds because the loss function $f(w, z)$, and thus the function $F(w)$, is $L$-Lipschitz. We next note that $\bar{A}_c$ computes the point $\widetilde{w}$ by running the algorithm $\bar{A}_{sc}$ with inputs $U, \widehat{w}$ on the loss function $\widehat{f}$. Thus, we have that

$$
\mathbb{E}[\|\widetilde{w} - \widehat{w}\|] = \mathbb{E}\left[\left\|\frac{1}{n - m}(\widehat{H})^{-1} \sum_{z \in U} \nabla \widetilde{f}(\widehat{w}, z) + \nu\right\|\right],
$$

where $\widehat{H} := \frac{1}{n-m} \sum_{z \in S \setminus U} \nabla^2 \widetilde{f}(w, z)$. Since, the function $\widetilde{f}$ is $\lambda$-strongly convex, we note that $\widehat{H} \succcurlyeq \lambda \mathbb{I}_d$. Using this fact with the above relation implies

$$
\begin{aligned}
\mathbb{E}[\|\widetilde{w} - \widehat{w}\|] &\leq \frac{1}{\lambda(n - m)} \|\sum_{z \in U} \nabla \widetilde{f}(\widehat{w}, z)\| + \mathbb{E}\|\nu\| \\
&\leq \frac{1}{\lambda(n - m)} \sum_{z \in U} \|\nabla \widetilde{f}(\widehat{w}, z)\| + \mathbb{E}\|\nu\| \leq \frac{2mL}{\lambda(n - m)} + \sigma,
\end{aligned}
\tag{27}
$$

where the last line follows from the fact that $\|\widehat{w}\| \leq \frac{L}{\lambda}$, and thus

$$
\|\nabla \widetilde{f}(\widehat{w}, z)\| \leq \|\nabla f(\widehat{w}, z)\| + \lambda \|\widehat{w}\| \leq 2L.
$$

Finally, using (27) and Lemma 11 in (26), and by plugging in the value of $\sigma$ implies the desired performance guarantee. $\square$

Finally, the algorithms $A_c$ and $\bar{A}_c$ are $(\varepsilon, \delta)$-forgetting, as a consequence of Lemma 10.

**Lemma 13** (Forgetting guarantee). *For any distribution $\mathcal{D}$, dataset $S$, set of delete requests $U \subseteq S$, the algorithms $A_c$ and $\bar{A}_c$ satisfy the following guarantees for any set $W \subseteq \mathbb{R}^d$,*

(a) $\Pr(\bar{A}_c(U, A_c(S), T(S)) \in W) \leq e^\varepsilon \Pr(\bar{A}_c(\emptyset, A_c(S \setminus U), T(S \setminus U)) \in W) + \delta$, and

(b) $\Pr(\bar{A}_c(\emptyset, A_c(S \setminus U), T(S \setminus U)) \in W) \leq e^\varepsilon \Pr(\bar{A}_c(U, A_c(S), T(S)) \in W) + \delta$.

**Proof of Theorem 3.** The desired statements follow immediately from Lemma 11, Lemma 12 and Lemma 13 respectively. $\square$

## D.2 Supporting technical results

The following lemma gives a bound on the regularized empirical risk minimizer point for the loss function $f(w, z)$ for any dataset $S$. This gives us a bound on the domain of interest, and thus allows us to bound the Lipschitz constant for loss function over this domain.

**Lemma 14.** *Let $f(w, z)$ be a $L$-Lipschitz function in the variable $w$, and let $\widetilde{f}$ be defined as*

$$
\widetilde{f}(w, z) = f(w, z) + \frac{\lambda}{2}\|w\|^2.
$$

*Given a dataset $S = \{z_i\}_{i=1}^n$, define $\widehat{G}(w) = \frac{1}{n} \sum_{i=1}^n \widetilde{f}(w, z_i)$, and let $\widehat{w}$ denote the empirical risk minimizer of the loss function $\widetilde{f}$ on dataset $S$, i.e. $\widehat{w} \in \operatorname{argmin}_w \widehat{G}(w)$. Then, the point $\widehat{w}$ satisfies $\|\widehat{w}\| \leq \frac{L}{\lambda}$.*

**Proof.** Since, $\widehat{w} \in \operatorname{argmin}_w \widehat{G}(w)$, we have

$$
\nabla \widehat{G}(\widehat{w}) = \frac{1}{n} \sum_{z \in S} \nabla \widetilde{f}(\widehat{w}, z) = 0.
$$

Plugging in the definition of the function $\widetilde{f}$ in the above implies that

$$\frac{1}{n} \sum_{z \in S} \nabla f(\widehat{w}, z) + \lambda \widehat{w} = 0.$$

Rearranging the terms, we get

$$\|\lambda \widehat{w}\| = \|\frac{1}{n} \sum_{z \in S} \nabla f(\widehat{w}, z)\| \overset{(i)}{\leq} \frac{1}{n} \sum_{z \in S} \|\nabla f(\widehat{w}, z)\| \overset{(ii)}{\leq} L,$$

where the inequality in $(i)$ follows from an application of the Triangle inequality, and the inequality in $(ii)$ holds because the function $f$ is $L$-Lipschitz in the variable $w$, and thus $\|\nabla f(w, z)\| \leq L$ for all $w \in \mathcal{W}$ and $z \in \mathcal{Z}$. $\qquad \square$