# OpenReview forum: "Remember What You Want to Forget: Algorithms for Machine Unlearning"
_NeurIPS.cc/2021/Conference — NeurIPS 2021 Poster_

### Official Review · Reviewer_vRcu · 2021-06-29

**Rating:** 6
**Confidence:** 2

**Summary:**

This paper studies the problem of unlearning: given a model M trained on a set of examples S, and a subset T of S, the goal is to generate a model M’ that does not rely on any of the samples in T. The authors in particular aim to minimize the test error (rather than the training errors is done in previous approaches), while keeping speed and space constraints low. Their main contributions are a comparison with differential privacy (DP) algorithms, and showing theoretically that unlearning and DP are two separate problems, as well as presenting an unlearning algorithm which can unlearn asymptotically more examples the DP.
I will say upfront that I am not an expert on this topic, but to the best of my judgement the paper seems interesting and solid.


**Limitations And Societal Impact:**

The paper tries to make models adhere to privacy requirements such as deletion of personal data, which makes it seem like it could have a positive societal impact.


**Main Review:**

The paper provides a thorough background and is generally clear. The contributions seem novel, though I am not intimately familiar with the topic.

Questions:
1. The authors attribute some of the limitations of previous approaches to their usage of randomness (#176), and later say explicitly that “(their) definition does not require the learning algorithm to be randomized”. However, as stated in #301 and Algorithm 1, their model in fact does add noise to w.
2. The authors discuss one of the benefits of their approach in not needing to save the entire training set. However, a reasonable assumption is that you don’t know in advance which samples are going to be deleted, so it seems like one would need to save the entire training set anyhow, plus a mapping from each author to their samples, which makes this constraint not a particularly useful one.

Minor:
#189: grown -> grow


**Time Spent Reviewing:**

2

---

> ### Author Response · Authors · 2021-08-10
> **Response to all your concerns**
>
> Thank you for your time and a detailed review. Please find response to all your concerns below. We would be happy to follow up further through the Openreview response system on any of the following if the reviewer wishes.
>
> &nbsp;
>
> 1. ***learning algorithm randomized?***
>
>    In our setup, we decouple the learning stage and the unlearning stage. Algorithm 1 that the reviewer is concerned about is actually the unlearning algorithm and is randomized. Our learning algorithm is deterministic and is provided Algorithm 2 on page 19 in the appendix.
>
>
> &nbsp;
>
> 2. ***Concern about the storage model***
>
>    In our setup, we assume that each unlearning request consists of the entire data sample description. Hence, we do not need to store them in the local memory of the unlearning algorithm.
>
>    &nbsp;
>
>    The above can be implemented as follows: Suppose that each data sample has a certificate that it was used during training. Hence, the delete request consists of the entire data sample description and the certificate of usage. In this case, the only additional resources needed are T(S).
>
>    &nbsp;
>
>    Another scenario of resource constraints where our model is useful is when the user data is stored in very slow to access persistent memory and is thus inconvenient to be accessed by learning / unlearning algorithms. In this case, we care about how much local memory (which is fast to access) is needed by the learning algorithm.

---

> > ### Comment · Reviewer_vRcu · 2021-08-11
> > **Thank you for the clarification**
> >
> > 2. I am not sure I follow. Are you suggesting that each delete request contains the *entire training set*? this seems like a rather inefficient solution, which would still require loading the data to memory. As to the second scenario, couldn't it be applied to other methods as well?

---

> > > ### Author Response · Authors · 2021-08-12
> > > **Illustrative example on the information provided to the unlearning algorithm**
> > >
> > > > Are you suggesting that each delete request contains the entire training set? this seems like a rather inefficient solution, which would still require loading the data to memory. As to the second scenario, couldn't it be applied to other methods as well?
> > >
> > > We apologize for the confusion. We do not assume that each delete request contains the entire training set; that would defeat the purpose of our algorithm altogether. *We only assume that the samples to be deleted  are mentioned in the deletion request*.
> > >
> > > Let us clarify with a toy example: Suppose we train a model $\widehat{w}$ with four examples $S = \{[0.0, 0.1], [2.0, 2.3], [4.0, 0.5], [3.0, 1.3]\}$. After training the model, the server only keeps $\widehat{w}$ and a cheap-to-store data statistic $T(S)$, and deletes $S$ from the memory. At this point of time the server doesn't have access to any point from $S$ anymore. Now, when a delete request comes, the request contains just the sample to be deleted e.g., $[4.0, 0.5]$, and does not contain the entire dataset $S$. The server unlearns the point $[4.0, 0.5]$ using the information $\widehat{w}$, $T(S)$, and $[4.0, 0.5]$ *only*  (and does not load or have access to $S$). Here, the data sample $[4.0, 0.5]$ is provided to the unlearning algorithm by the user who owns this data and requests for deletion.
> > >
> > > We hope that the above illustrative example resolves the concerns of the reviewer. Please let us know if you would like us to elaborate further on the above, or on any other aspect of our setup in a follow up response. Thanks!

---

> > > > ### Comment · Reviewer_vRcu · 2021-08-15
> > > > **Thank you for the detailed explanation**
> > > >
> > > > I understand the procedure now, thank you. As mentioned in my review, I am not an expert in this topic and perhaps not the typical reader of this paper, but it might make sense to add a similar example to the paper to illustrate the procedure.

---

> > > > > ### Author Response · Authors · 2021-09-01
> > > > > **will do**
> > > > >
> > > > > Thanks, we will try to better illustrate this in the final paper.

---

### Official Review · Reviewer_FKp5 · 2021-07-16

**Rating:** 6
**Confidence:** 4

**Summary:**

The paper proposes a new unlearning algorithm that is provably memory and sample efficient in some cases. The paper provides sample complexity upper and lower bounds for test error given access to second order information of loss function.


**Limitations And Societal Impact:**

Have addressed appropriately.

**Main Review:**

Strength: - Nicely written, with detailed related work section. Theorems and definitions are well motivated and each of them is intuitively explained for readers not so familiar with the problem of machine unlearning and the general area. The paper adopts standard Mathematical notations which is nice, and they are consistently followed throughout the paper as well.
- Application of ideas from DP for unlearning is intuitive and interesting. The main idea is that if the future delete requests are manageable, then a deterministic procedure like the one proposed maybe sufficient under some technical conditions on the loss function of the learning algorithm. The separation result in terms of sample complexity between DP and unlearning is technically interesting.

Weakness:- For a paper that presents a machine (un)learning _algorithm_ as its main contribution, it does not contain any experiments, not even simulation. It seems like the algorithm could be implemented for reasonable size datasets since it only requires hessian-vector product -- this package comes to mind https://github.com/amirgholami/pyhessian
- Again, for a theory paper like this, I consider it a lost opportunity to give no intuition at all about the proofs. I believe readers would be able to appreciate the paper even more if some proof sketches are provided. The examples are nice but they are almost trivial, and they don't necessarily convey the challenge . The proofs contain some minor typos and quantities (D^*) undefined. The proofs appear to be technically correct and utilize tools from convex optimization and results from differential privacy literature.

Justification: The paper seems to have some interesting ideas that are not fully formed i.e., pieces of the paper sometimes felt disjoint without one full coherent story. For example, the introduction section is only slightly related to the rest of the paper -- discussion about regulations is totally forgotten. A similar statement could be made for the related work section, wherein it seems none of the ideas are used in this paper. Incorporating some elements of the proofs in the main paper might make it more readable from the technical perspective.


**Time Spent Reviewing:**

5

---

> ### Author Response · Authors · 2021-08-10
> **Response to all your concerns**
>
> Thank you for your time and a detailed review. Please find below our response to your concerns.
>
> ***Experiments***
>
> The primary focus of our work is theoretical investigation of machine unlearning with resource constraints. On the experimental side, there is already some evidence on the quality of our unlearning algorithm from the work of Guo et. al. 2020. They considered a similar update rule (see lines 103-112 in our paper for a discussion on the differences between the two works) and performed experiments on linear models based on pre-trained neural network features and on logistic regression. Aligning with our theoretical conclusions, their experiments demonstrate that the provided unlearning rule retains good performance even after deleting a significant fraction of the training data. We will add a pointer to their empirical observations in the final version of the paper.
>
> ***Paper presentation***
>
> We thank the reviewer for suggestions to improve the paper. We will add more technical intuition and clarify our setup further in the final version of the paper, which we could not do in the submission draft due to space constraints. We believe the full version of the paper (submitted as supplement) has the most cohesive version of our story, but sadly parts of it might have been lost in translation to the proceedings version. Any more concrete suggestions towards improvement would be greatly appreciated.

---

> > ### Author Response · Authors · 2021-08-22
> > **Follow up on our response**
> >
> > Respected Reviewer,
> >
> > Please let us know if our response addresses your concerns. We would be happy to engage in additional discussion through the Openreview response system to clarify things further, if the reviewer wishes.
> >
> > Thanks!

---

> > > ### Comment · Reviewer_FKp5 · 2021-08-25
> > > **Thanks for response**
> > >
> > > Thanks for providing the clarifications, I have raised my score accordingly.

---

### Official Review · Reviewer_yPiR · 2021-07-17

**Rating:** 6
**Confidence:** 4

**Summary:**

The paper considers the problem of machine unlearning: after a model is trained on a dataset, there is a request to delete a point from the dataset. The goal is to design an *efficient* method to update the trained model such that is nearly indistinguishable from what we would have obtained had we trained on the dataset without the point. The authors use differential privacy as the unlearning criteria and look at unlearning in convex learning problems. The give a method which can *unlearn more points* that DP-training with small runtime and storage complexity.

**Limitations And Societal Impact:**

No direct societal impact.

**Main Review:**

The paper contributes novel definitions and terminology to the budding unlearning literature. To give an example: the viewpoint of deletion capacity is interesting.
The paper is well-written, the proofs are natural and well-presented.

**Confusion with the subtlety in the definition of unlearning**: The authors claim that their definition of $(\epsilon, \delta)$ unlearning is more general than previous work because it allows the learning algorithms to be deterministic. But I feel that this is an dubious claim from a practical viewpoint basically because any deterministic output is useless to begin with. The promise of unlearning is that the updated model is indistinguishable from what is obtained from training on the dataset with the deleted points. If the model in use is the deterministic one, then the company/service is not compliant  with the unlearning promise to begin with, and hence the deterministic output serves no purpose. The only situation where the deterministic output is thus useful is, when we know there are no unlearning requests to address  -- no reason to believe why we know this a-priori.

**Separation between DP and unlearning**:
A key contribution in the paper is a separation of DP and unlearning. It should be noted though that unlearning algorithms operate under more information than DP (they have knowledge of the deletion requests), therefore a smaller accuracy is reasonable to expect -- showing this formally is a good contribution. However, upon thinking about it a little more, it semms to me that it is possible to show much stronger results (see later half of "population risk minimization and non i.i.d. points" paragraph).

**Population risk minimization and non i.i.d. points**:
This paper, in contrast to most previous work, considers excess population risk (rather than excess empirical risk) as the measure of performance. The authors motivate this via an example of mean estimation where it is straightforward to unlearn with best empirical risk performance. However, the authors show that there exists a sequence of deletions such that the corresponding population risk deteriorates. Interestingly, the deletion sequence is "adversarial", in the sense that the data points are no longer i.i.d., and thus (at a high-level) the empirical minimizer fails. My first impression upon reading this was that this is a challenging problem, since if we allow such "adversarial" updates, then we need to devise learning algorithms for datasets which are not i.i.d. anymore. But (if my understanding is correct) initial i.i.d. points and stability properties of ERM allows them to tackle it. My understanding is as follows: the initial dataset is i.i.d. and we have a model (empirical minimizer) $w_1$ which has small population risk. After deletions, we have to output $w_2$, the new empirical minimizer. It can be shown that $\Vert w_1-w_2 \Vert \leq \alpha$, so with the DP-like unlearning guarantee, we can add noise $\propto \alpha$, but this leads to worse accuracy. Instead, is there a point $w_3$, which is *closer* to $w_2$ but still has good population risk performance (guaranteed if it is close to $w_1$) -- the proposed unlearning algorithm gives such a point *efficiently*.
The two factors described above gives the two terms in the accuracy guarantee.
Now, ignoring the computational cost of unlearning, I think that there are *even better* models, ones which optimize the aforementioned trade-off optimally. Consider a model $w_4$: let $a:=\Vert w_1 - w_4\Vert $ and $b:=\Vert w_2 - w_4\Vert $.  Let $c:=\Vert w_1 - w_2\Vert \leq \frac{m}{\lambda n}$. If $w_1, w_2$ and $w_4$ are co-linear, then $a+b=c$. Adding noise of variance (roughly) $b^2/\epsilon$ suffices for the unlearning guarantee. Plugging this in the proof of Lemma $9$, this gives an excess risk of $a + \sqrt{d}b/\epsilon + \frac{1}{\lambda n}$, the trade-off between the first two is optimized when $a = \frac{\sqrt{d}}{\epsilon+\sqrt{d}}c \leq c$. This gives an excess risk of $\frac{m}{\lambda n}$.
Note that this is as good (in terms of upper bounds) as using $w_2$ with no noise.
It is unclear if we can find this $w_4$ efficiently, but the above argument (if correct) is sufficient to establish a much stronger separation between unlearning and DP. Please correct if I made any mistake in the above reasoning.

**Population risk vs ERM**: While the authors make a big deal about the fact that unlike prior work, they consider the population risk problem, in their problem (strongly convex learning), simply outputting (approximate) ERM gives smaller or (at worst) equal excess population risk than than authors'. Importantly, this is the method and proof technique used by the authors.

**Some confusion on storage**:
A key criteria in the paper is to also minimize storage complexity.
The authors want that storage complexity be independent of $n$, and moreover, the dataset is not stored/accessible. My confusion is, when getting a point $z$ to deleted, do we get (a). an id of the point or (b). full description of the point. If it is (a), then we indeed need to store the whole dataset, since we don't know the incoming requests. If it is (b), then the storage is indeed independent of $n$, but then what are we even "remembering" (which motivates the paper title). In general, the phrase "remember what you want to forget" is confusing to me -- why is it fine to assume we know what we want to forget a-priori?

**Unlearning runtime**:
The authors say that their unlearning runtime is $O(d^\omega)$, however, they should also consider the time it takes to compute the hessian and gradient on the deleted data points -- the sum of Hessians overl all data-points can be be pre-computed, but not the individual terms: this gives additional runtime of $\Omega(md^2)$.

**Comparison with retraining**:
A natural baseline in unlearning papers is comparison with retraining.
With the constraints put on storage, the authors specifically removed retraining as a valid method.
However, I find this limiting since in such a new and unexplored problem as unlearning, having multiple criteria of performance (unlearning runtime, storage..)  inhibits a deep understanding of any of the aspects. Furthermore, if we remove the only known (and trivial) method, then how to do we if the guarantees are indeed *interesting* (perhaps the trivial method retraining takes space but has much *smaller* runtime?).
Thus, due to personal curiosity, I tried to (roughly) calculate how it performs against retraining.
Due to the population risk minimization criteria and non-i.i.d. points, I first thought that this does not apply.
However, due to the stability properties of ERM (the approach via which this paper gave risk bounds), the ERM output after $m$ deletions can be shown to have excess risk $\frac{m}{\lambda n}$ -- the authors' upper bound is, at best, this term. Similarly, an $\alpha$ approximate ERM can be shown to have excess population risk of $\sqrt{\frac{\alpha}{\lambda}} + \frac{m}{\lambda n}$.
The authors achieve excess population risk of $\frac{\sqrt{d}m^2}{\lambda^3 n^2 \epsilon}$ (disregarding other terms) with run-time $md^2 + d^\omega$. To achieve the same error on $n-m = \Omega(n)$ points, the total gradient complexity (using SGD) is $\frac{\lambda^4 n^4 \epsilon^2}{d m^4}$. Hence, the prposed method is more efficient than retraining if (roughly) $\max(md^2, d^{\omega}) < \frac{\lambda^4 n^4 \epsilon^2}{d m^4} \iff \epsilon > \frac{m^2}{\lambda^2n^2}  \max \left( \sqrt{m}d^{3/2}, d^{(1+\omega)/2}\right)$. For reasonable unlearning guarantee, we want $\epsilon$ to be a small constant, say $1$. This would give (reasonable?) upper bounds on $m$ and/or $d$ in terms of number of data points $n$ and $\lambda$.
Again, I think such an exercise as above is required to assess if the guarantees are indeed non-trivial.

**Incorrect description of previous work:** The authors mention "Ullah et al. 2021, Machine Unlearning via Algorithmic Stability" in context of methods which use an approximate unlearning criteria (i.e. differential privacy). However, this is incorrect: Ullah et al. 2021 give methods which guarantee **exact** unlearning.


**Time Spent Reviewing:**

6

---

> ### Author Response · Authors · 2021-08-10
> **Response to all your concerns**
>
> Thank you for your time and for a detailed review. Please find below our response to all your concerns. *We would be happy to further follow up on any of your concerns through the openreview response system if the reviewer wishes.*
>
> 1. ***Confusion with the subtlety in the definition of unlearning:***
>    *  In our setup, we decouple the learning stage and the unlearning stage. We consider a three stage process: (1) First we learn a model on the dataset $S$ and deploy it, (2) there is a call for unlearning requests and data samples $U$ request to be deleted, (3) we update the model to  acknowledge these unlearning requests.
>
>    &nbsp;
>
>    * Our claim stated in the paper is that the learning algorithm deployed in stage (1) (and before stage (2)) does not have any additional randomness. We believe the reviewer may have interpreted the claim to be about the entire learning-unlearning pipeline, and we will clarify this point in the final paper.
>         We only add noise in stage (3), when acknowledging the deletion requests. Note that, our unlearning algorithm can choose to add noise for stage (3) even when there are no delete requests in stage (2).
>
>       &nbsp;
>
>       *Advantage of this model:* Since no noise is added until the first unlearning request comes in stage (2), the performance of the learning algorithm does not degrade in stage (1).
>
>       &nbsp;
>
>    Finally, we note that our unlearning definition is more general than the definitions considered in prior work as
>
>       (a) Our definition also allows randomized learning algorithms (if needed) and does not restrict to deterministic learning algorithms. However, for our upper bounds, considering deterministic learning algorithms suffices.
>
>       &nbsp;
>
>       (b) Our definition also allows direct comparison to the output of a randomized learning algorithms by explicitly defining $\bar{A}$ to add noise when there are no delete requests; hence recovering the unlearning definition in the prior work.
>
> &nbsp;
>
> 2. ***Separation between DP and unlearning:***
>
>    Our key contribution in this paper is to theoretically demonstrate this separation between DP and machine unlearning under resource constraints, as the reviewer's intuition suggests. We note that such a separation was not known before.
>
> &nbsp;
>
> 3. ***Population risk minimization and non i.i.d. points:***
>
>    The reviewer existentially argues that there may be a stronger separation between unlearning and DP, by demonstrating existence of the point $w_4$ which is better than the point $w_3$ output by our algorithm in terms of test loss performance.
>       * If one simply wanted to show a stronger separation, then one could simply consider the point $w_2$. This satisfies perfect unlearning, and is clearly better than $w_3$, $w_4$, or indeed, any other possible point.
>       * However, as the reviewer mentions the crucial point, it is not clear how to compute either $w_2$ or $w_4$ subject to memory or computational constraints.
>
>    &nbsp;
>
>    Therefore, in the setting where we are subject to resource constraints, our separation result is novel. Without these constraints, one could simply retrain from scratch to obtain $w_2$; hence giving a stronger separation.
>
>    &nbsp;
>
>    However, demonstrating a stronger separation between unlearning and DP by showing existence of a better point is a very interesting research direction, and we will think more along this direction. We thank the reviewer for bringing this to our attention.
>
> &nbsp;
>
> 4. ***Population risk vs ERM:***
>
>    First, we argue that in general, considering the empirical loss as the objective may not be the right thing to do in machine unlearning. In fact, ERM is not the correct objective even for the simpler problem setting of machine learning (as shown in [Shalev-Shwartz et al., 2009a, Feldman, 2016] for stochastic convex optimization and more recently in [Kale et al. 2021] for regularized objectives). However, for certain problems with additional structure it may still be the case that minimizing the empirical loss suffices for minimizing the population loss (as in strongly convex optimization); but this recipe is wrong in general.
>
>    &nbsp;
>
>    We thus consider the reframing of the problem in terms of population risk to be important in itself. Indeed, this reframing is what allows us to demonstrate a separation between DP and unlearning, which is our second contribution. We agree that for strongly-convex loss functions, the prior work that minimizes empirical loss also has implications for population loss, though it is not explicitly analyzed as such. Additionally, the prior work does not consider storage constraints, or discuss this separation between DP and unlearning.
>
> &nbsp;
>
> 5. ***Some confusion on storage:***
>
>    In our model, we assume that during unlearning, we get the full description of the data sample to be unlearnt. Thus, we do not store the training data in the local memory available to the learning / unlearning algorithms. User data may be stored by the organization at some other location where accesses may be costly, or restricted to specific circumstances (such as deletion requests) due to privacy concerns.
>
>    &nbsp;
>
>    We like the name for two reasons: First, we need to store (or “remember”) some additional data statistics during the learning stage so as to implement our unlearning algorithm. Second, during the unlearning stage, we can not ignore the data points that want to be deleted, or else we fall back to DP based methods.
>
> &nbsp;
>
> 6. ***Unlearning runtime:*** Thanks for pointing out this typo. We will fix this.
>
> &nbsp;
>
> 7. ***Comparison with retraining***
>
>     Our focus in this paper is to efficiently unlearn under memory constraints, and not to develop the fastest algorithm for unlearning. The latter has been the focus of many prior works in machine unlearning (as we discussed in Section 1.1). Hence, while we agree that there are some regimes with respect to running time where retraining from scratch could be faster than computing the hessian and its inversion (which we require for our unlearning algorithm),
> retraining from scratch requires storing the entire dataset, and is simply not feasible under memory constraints. In this case, the simplest baseline algorithm one can compare to is unlearning methods that directly use differential privacy, which we compare to in the paper.
>
> &nbsp;
>
> 8. ***Incorrect description of previous work:*** Thanks for pointing this out. We will fix this!

---

> > ### Comment · Reviewer_yPiR · 2021-08-16
> > **Follow-up**
> >
> > I thank the authors for their clarifications. Some confusions that remain are:
> >
> > 1. Definition of unlearning:
> >
> > Perhaps not an important concern, but I am still confused as to why the author's notion is an appropriate notion of unlearning. Specifically, say model "w = A(S)" is deployed after stage 1, then after stage 3, the updated model is w'. A reasonable criteria would be that w' and A(S\ U) are close, but here, the guarantee is that w' is close to A'(S), which is noisy A(S). If my understanding is correct, then why is the above criteria reasonable?
> >
> > 2. Separation between DP and unlearning/ Population risk and non-iid points:
> >
> > Thanks for the clarification. It seems to me that "Separation between DP and unlearning" is perhaps not the best phrase and which is what misled me --  my understanding of this was a separation between DP algorithms = no knowledge of unlearning requests and unlearning algorithms = complete knowledge of unlearning requests. With this, as the authors' pointed out, a separation result is trivial: just output the ERM on the updated dataset. The authors setting instead additionally include the constraint that memory is *small* (full dataset cannot be stored), which discards ERM as an unlearning algorithm. I would suggest that the authors provide a formal description of the model under which this separation result is obtained.
> >
> > 3. Population risk vs ERM:
> >
> > While what the authors say is important and interesting, my high-level point here is that some of the build-up to the main result feels misleading. Specifically, the mean estimation example, rather disconnected from the rest of the paper, motivates why ERM, though easy to unlearn, is not the right thing to do for the population risk problem. However, in the main result, the algorithm used for the population risk problem is ERM.

---

> > > ### Author Response · Authors · 2021-08-20
> > > **Response to follow-up questions**
> > >
> > > Thank you for further clarification questions. Please find our response below.
> > >
> > > * **Definition of unlearning**
> > >
> > >     Let $U$ be the unlearning requests. We compare our deployed model in stage-3 to the corresponding model we would have deployed in stage-3 in an imaginary world where U was not in the original dataset (and correspondingly no point wanted to be unlearnt). Note that the comparison between the two worlds is made at stage-3 only, **after the call** for unlearning requests in stage-2.
> > >
> > >    &nbsp;
> > >
> > >     Our unlearning definition allows for the model deployed in stage-1 to depend on all the data points. In fact, the model deployed in stage-1 does not need to satisfy any privacy guarantees (as no body wanted to be unlearnt yet).
> > >
> > >   &nbsp;
> > >
> > >
> > >     In addition to the above, our definition also allows direct comparison to $A(S \setminus U)$ by restricting to unlearning algorithms $\bar{A}$ for which $\bar{A}(A(S \setminus U), \emptyset) = A(S \setminus U)$ + noise. Such algorithms can be easily implemented by modifying $\bar{A}$ to simply return the input model $A(S \setminus U)$ + noise when the set of delete requests is $\emptyset$.
> > >
> > > &nbsp;
> > >
> > > * **Separation between DP and unlearning/ Population risk and non-iid points**
> > >
> > >    Apologies if our response caused confusion. As the reviewer mentioned, our separation result is indeed between DP = no knowledge of unlearning requests, and unlearning algorithms = complete knowledge of unlearning requests.
> > >
> > >    &nbsp;
> > >
> > >    However note that one cannot "just output the ERM on the updated dataset" as the reviewer mentions. This is due to the fact that throughout the paper, we consider the unlearning setup with resource constraints. Hence, the original dataset / updated dataset are not available during the unlearning phase. Therefore showing a separation requires additional work, which we do in Section 3.2 and 3.3.
> > >
> > >     Hope this clarifies the concern.
> > >
> > >  &nbsp;
> > >
> > > * **Population risk vs ERM**
> > >
> > >    There are two main contributions in the paper: (a) Arguing that ERM is not the right objective to look at, in general, for machine unlearning, and (b) To show that under resource constraints, there exists algorithms which can be better than simply using DP.
> > >
> > >    &nbsp;
> > >
> > >
> > >    Our example of convex loss functions was to show part-(b). For part-(a), as also shown by e.g. Shalev-Shwartz et al, 2009a, Feldman, 2016 and Kale et al. 2021, points that minimize the empirical loss may not get good test loss performance in general.  We will clarify this further in the paper by discussing these examples as well, and their  consequences for machine unlearning.
> > >
> > >    &nbsp;
> > >
> > >
> > >    The contributions (a) and (b) are focusing on different aspects of machine unlearning, and we request the reviewer to consider them separately.
> > >
> > > &nbsp;
> > >
> > > Please let us know if the reviewer wishes to discuss further on any of these concerns. We would be happy to follow up through the Openreview response system.

---

> > > > ### Comment · Reviewer_yPiR · 2021-08-20
> > > > **Repsonse**
> > > >
> > > > I thank the authors for their reponse -- this has been very helpful to me to understand some of the details better; hopefully, it was useful to the authors too. As a result, I pan to increase my score for the paper. Some final suggestions:
> > > > 1. Modify the part where mean estimation is introduced to motivate the population risk problem - as I said before, it was confusing to me since the main training algorithm in the paper is ERM. I hope that a polished disucssion with added references, which the authors mention above, will help the highlight the key message here.
> > > > 2. When mentioning the separation result, define (informally) class of algorithms: DP and unlearning, and mention the resource constraint therein. This resource constraint is what makes the separation result novel.

---

> > > > > ### Author Response · Authors · 2021-08-22
> > > > > **Thank you!**
> > > > >
> > > > > Thank you reviewer for bringing these points forward, and for spending their time in carefully reading and discussing the paper with us.
> > > > >
> > > > > In addition to the changes that we promised in our earlier response, we will be sure to also add both of the requested discussion points in the final version of the paper.

---

### Official Review · Reviewer_AJse · 2021-07-18

**Rating:** 7
**Confidence:** 4

**Summary:**

Machine unlearning is new and hot topic. This paper makes several interesting contributions. The idea that the unlearning operation can only access a summary of the entire dataset is a great conceptual innovation since many deployment environments are memory-constrained compared to the training environment. The main contributions of this paper lean theoretical. I did not closely check all the details in the math, but it seems correct at a high-level. I consider the theoretical contributions interesting and strong.

**Limitations And Societal Impact:**

Yes, they address social impact.

**Main Review:**

Contributions:
 - Study trade-off between generalization loss and unlearning with memory constraints
 - Propose an unlearning technique and show it results in asymptotic improvements over DP-based unlearning methods

Overall, I am enthusiastic about this paper. I am tentatively voting a 7.

Suggestions/Concerns:

- One of the key, interesting findings of this paper is the strict separation between DP and removal in the setting of choice. I suggest the authors add more intuition and description so that the readers can easily digest this finding.

- The deletion notion in Ginart 2019 does not require randomness, contrary to the claims in this paper. In that work, the learning algorithm could just deterministically output the database in the clear and the unlearning operation need just remove the target row from the database. Even simpler, all learning algorithms could just deterministically output "0" and the unlearning operation doesn't need to do anything. The only differences I can see with respect to the definition in Ginart 2019 are that: (1) this definition explicitly supports batch deletions, (2) this definition is two-sided, meaning that unlearned models must be indistinguishable from models trained on the subset  & (3) this definition limits access to the dataset via the summary structure T.

- Consider letting the magic constant (0.01) be an arbitrary gamma. For some applications 0.01 is too small and for others it might be too large. This is a stylistic recommendation and does not need to be addressed in rebuttal.

- While this paper clearly leans towards the theoretical, it would be good to include simple experiments to illustrate the points made in the paper. For example, a simple linear or logistic regression setting should satisfy the theoretical criteria and is not a hard implementation. It is nice to actually compute and tell the reader what the actual improvement in deletion capacity ends up being, especially if non-vacuous metrics can be achieved in practice.

- At the end of Section 4, the subsection titled "Algorithms for convex loss functions" is elementary and should be moved to the appendix. It is standard knowledge that adding regularization to the loss function yields strong convexity and this does not need an entire subsection to explain (it can be stated in 1 or 2 sentences as above, if mentioned at all).


**Time Spent Reviewing:**

3

---

> ### Author Response · Authors · 2021-08-10
> **Response**
>
> Thanks for the detailed and positive review. We will incorporate all of your feedback in the final version of the paper, and add more discussion as you recommended. Please find below response to your concerns:
>
> * ***"Deletion notion in Ginart 2019..."***
>
>    Thanks for pointing this out. We will add a discussion on this in the final version of the paper.
>
> &nbsp;
>
> * ***Experiments***
>
>    The primary focus of our work is theoretical investigation of machine unlearning with resource constraints. On the experimental side, there is already some evidence on the quality of our unlearning algorithm from the work of Guo et. al. 2020. They considered a similar update rule (see lines 103-112 in our paper for a discussion on the differences between the two works) and performed experiments on linear models based on pre-trained neural network features and on logistic regression. Aligning with our theoretical conclusions, their experiments demonstrate that the provided unlearning rule retains good performance even after deleting a significant fraction of the training data. We will add a pointer to their empirical observations in the final version of the paper.

---

### Decision · Program_Chairs · 2021-09-27

**Decision:**

Accept (Poster)

**Comment:**

The paper considers the problem of machine unlearning: after a model is trained on a dataset, there is a request to delete a point from the dataset. The goal is to design an efficient method to update the trained model s.t. is nearly indistinguishable from what we would have obtained had we trained on the dataset without the point. One baseline from prior work is to use DP. The authors give a method that can efficiently unlearn more points that DP-training in the context of in convex learning problems. The reviewers agree that this is an interesting paper with solid contribution, and all of them support acceptance.